# Asymmetric dinitrogen-coordinated nickel single-atomic sites for efficient $CO_2$ electroreduction

Yuzhu Zhou[1,4], Quan Zhou[1,4], Hengjie Liu ◉[1,4], Wenjie Xu[1], Zhouxin Wang[1], Sicong Qiao[1], Honghe Ding[1], Dongliang Chen[2], Junfa Zhu ◉[1], Zeming Qi[1], Xiaojun Wu ◉[3], Qun He ◉[1] ✉ & Li Song ◉[1] ✉

Developing highly efficient, selective and low-overpotential electrocatalysts for carbon dioxide ($CO_2$) reduction is crucial. This study reports an efficient Ni single-atom catalyst coordinated with pyrrolic nitrogen and pyridinic nitrogen for $CO_2$ reduction to carbon monoxide (CO). In flow cell experiments, the catalyst achieves a CO partial current density of 20.1 mA $cm_{geo}^{-2}$ at −0.15 V vs. reversible hydrogen electrode ($V_{RHE}$). It exhibits a high turnover frequency of over 274,000 $site^{-1} h^{-1}$ at −1.0 $V_{RHE}$ and maintains high Faradaic efficiency of CO ($FE_{CO}$) exceeding 90% within −0.15 to −0.9 $V_{RHE}$. Operando synchrotron-based infrared and X-ray absorption spectra, and theoretical calculations reveal that mono CO-adsorbed Ni single sites formed during electrochemical processes contribute to the balance between key intermediates formation and CO desorption, providing insights into the catalyst's origin of catalytic activity. Overall, this work presents a Ni single-atom catalyst with good selectivity and activity for $CO_2$ reduction while shedding light on its underlying mechanism.

Electrochemical reduction of carbon dioxide ($CO_2$) to carbon-based fuels and chemicals is a crucial part of the Carbon Capture, Utilization, and Storage (CCUS) technology, with broad application prospects in renewable energy storage and negative $CO_2$ emission[1–3]. The development of efficient electrocatalysts for $CO_2$ reduction is a research frontier, and a desirable $CO_2$ reduction reaction ($CO_2$RR) electrocatalyst should achieve low overpotential and high current density to products, while maintaining high selectivity and activity retention after long-term electrolysis[4–9]. Unfortunately, achieving ideal performance of $CO_2$RR catalysts is challenging due to the strong chemical bonding strength of $CO_2$ (806 kJ $mol^{-1}$) and competing hydrogen evolution reaction[10,11]. Although many efforts have been made over the past decade to develop ideal catalysts, most reported electrocatalysts exhibit high overpotentials and slow reaction rates, which limit their energy efficiency to a low level[12–15]. While some noble metal catalysts

like gold (Au) and silver (Ag) with high carbon monoxide (CO) selectivity can achieve low overpotentials in aqueous solution, their high cost and insufficient current density limit practical operation[16,17]. In contract, 3d transition metal-nitrogen-carbon catalysts (TM-N-C) such as Fe, Co, or Ni, offer a promising alternative to noble metals for electrochemical $CO_2$-to-CO conversion due to their low price and reserve abundance[18–20]. Both experimental and theoretical studies have evidenced that TM-N-C have good $CO_2$ activation and weak hydrogen binding capacities, resulting in good $CO_2$RR performance[20,21]. However, these catalysts generally exhibit much higher overpotentials than Au and Ag, limiting reaction efficiency[22–26]. The difficulty of improving current density at very negative potentials due to the desorption of CO on the TM sites is the main reason[16]. Hence, it is necessary and challenging to establish earth-abundant 3d metal-based catalysts with both low overpotential and high current

[1]National Synchrotron Radiation Laboratory, CAS Center for Excellence in Nanoscience, University of Science and Technology of China, Hefei 230029, China. [2]Beijing Synchrotron Radiation Facility, Institute of High Energy Physics, Chinese Academy of Sciences, Beijing 100049, China. [3]Hefei National Laboratory for Physical Science at the Microscale, Collaborative Innovation of Center of Chemistry for Energy Materials (iChEM), School of Chemistry and Materials Sciences, University of Science and Technology of China, Hefei 230026, China. [4]These authors contributed equally: Yuzhu Zhou, Quan Zhou, Hengjie Liu. ✉e-mail: hqun@ustc.edu.cn; song2012@ustc.edu.cn

density to substitute noble metals for the catalysis of CO-selective $CO_2RR$.

To address the above problem, this study presents an asymmetric dinitrogen-coordinated Ni single-atom catalyst (Ni-N-C). Electrochemical $CO_2RR$ tests in flow cell show that the obtained Ni-N-C can achieve great $CO_2$-to-CO conversion performance with a low overpotential of $-0.15$ V vs. reversible hydrogen electrode ($V_{RHE}$) at a CO partial current density of 20.1 mA $cm_{geo}^{-2}$, high Faradaic efficiency of CO ($FE_{CO}$) exceeding 90% within $-0.15$ to $-0.9$ $V_{RHE}$, high turnover frequency (TOF) of over 274,000 $site^{-1}$ $h^{-1}$ at $-1.0$ $V_{RHE}$, and decent stability for 60 h at a current density of ~450 mA $cm_{geo}^{-2}$. Meticulous characterization results and theoretical calculations suggest that single CO adsorption under electrochemical working conditions positively regulates the adsorption and activation of key intermediates on active sites, thereby achieving good $CO_2RR$ performance.

## Results and Discussion
### Structural identification of Ni-N-C

Ni-coordinated graphitic carbon nitride (g-$C_3N_4$) was thermally treated with argon and ammonia atmospheres, resulting in the formation of Ni-N-C and metal-free N-doped C (NC) as a control sample[27]. Ni phthalocyanine was also deposited on NC as a second control sample (NiPc)[28]. Transmission electron microscopy (TEM) and X-ray photoelectron spectra (XPS) analyses confirmed the obtainment of g-$C_3N_4$ based precursors, NC, and Ni-N-C (figs. S1 and S2). Inductively coupled plasma atomic emission spectroscopy (ICP-AES) analysis of Ni presence revealed the mass ratios of Ni in Ni-N-C and NiPc to be about 1.10 and 1.51 percent, respectively (table S1). X-ray diffraction (XRD) patterns of both Ni-N-C and NC showed two diffraction peaks at about 26° and 44°, corresponding to the (002) and (101) planes of the graphite arrays (fig. S3). No diffraction peaks associated with crystalline Ni were seen for either sample, indicating that the Ni species in Ni-N-C and NiPc

were highly dispersed or amorphous. The broadening of the diffraction peaks suggested the presence of defective carbon substrate, which was favorable to anchoring Ni atoms. Aberration-corrected high-angle annular dark-field scanning transmission electron microscopy (HAADF-STEM) confirmed that Ni species in Ni-N-C existed as isolated atoms (Fig. 1a), and elemental mapping analysis confirmed uniform distribution of Ni (Fig. 1b). The spectral intensity of Ni L-edge X-ray absorption near-edge structure (XANES) was weaker, and the $K_\beta$ peak of X-ray emission spectra (XES) was positively shifted for Ni-N-C compared to NiPc ($Ni^{2+}$), indicating the low-valence character of Ni (Fig. 1c and fig. S4). The above Ni K-edge XANES and XPS analyses have indicated that the oxidation state of Ni in Ni-N-C was less than +2. As shown in Fig. 1d, the absorption edge energy of Ni-N-C was lower than that of NiPc with $Ni^{2+}$. Because the absorption edge energy is positively correlated with the oxidation state, compared to NiPc, the detected absorption edge energy of Ni-N-C is lower, indicating that Ni sites in Ni-N-C have an average oxidation state of less than +2. The measured relatively low binding energy of Ni $3d$ in XPS spectrum of Ni-N-C further supports the above speculation (fig. S5). In addition, quantitative analysis of Ni oxidation states in Ni-N-C was achieved through the derivative analysis of XANES data with Ni foil and NiPc as references (fig. S6 and table S2). Quantitative absorption energy analysis suggests the energy values are 8333.00 eV, 8337.30 eV, 8339.28 eV for Ni foil, Ni-N-C, and NiPc, respectively. This demonstrates an average valence of +1.37 of Ni in Ni-N-C, demonstrating that Ni-N-C contains low-valence Ni sites with considerable proportion. Furthermore, the different characteristic features (I-IV) between Ni-N-C and NiPc suggested that they had different local coordination environments. The feature I, which resulted from the $3d-4p$ orbital hybridization forbidding electric dipole but allowing $1s-3d$ transition in the quadrupole, was stronger for Ni-N-C than for NiPc. This result indicated that the local coordination structure of Ni sites in Ni-N-C was highly disordered/

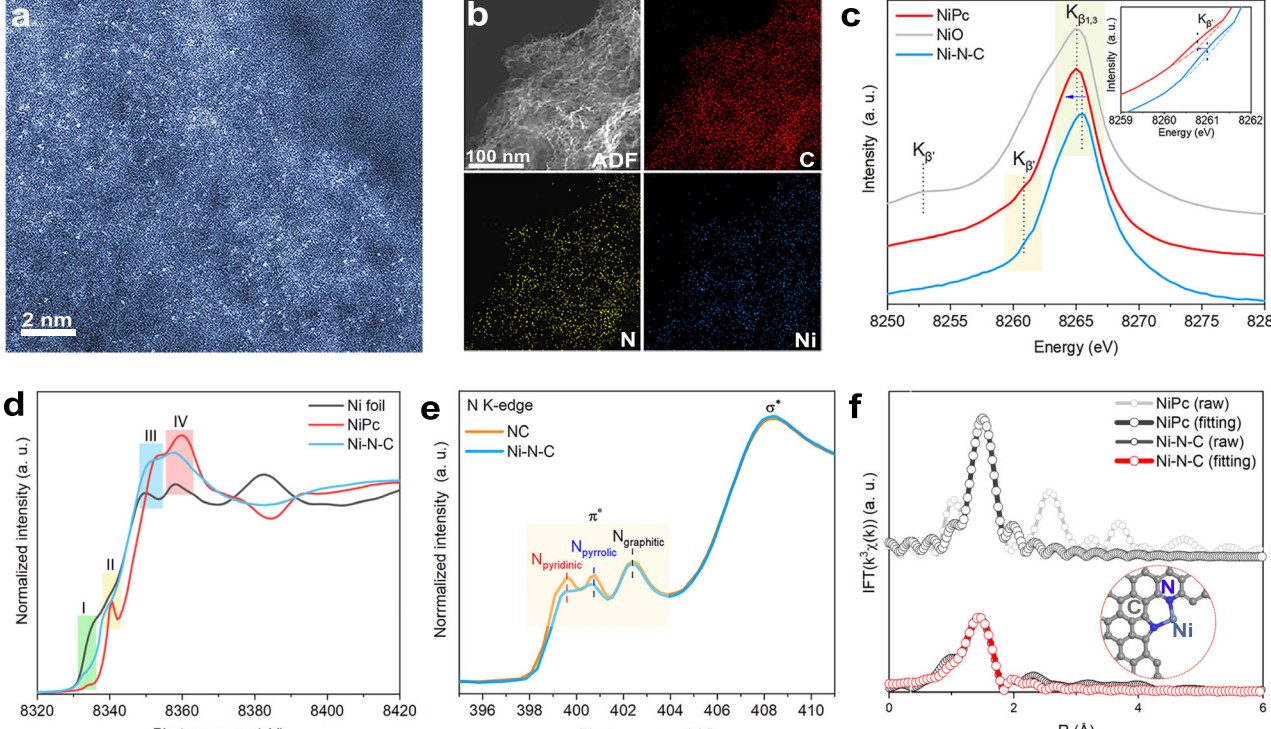

**Fig. 1 | Structural identification of Ni-N-C. a** Aberration-corrected HAADF-STEM image. **b** Elemental mapping. **c** XES spectra of Ni-N-C, NiPc, and NiO. Inset is the zoom in of $K_{\beta'}$ located at the yellow region. **d** Ni K-edge XANES spectra of Ni-N-C, Ni foil, and NiPc. **e** N K-edge XANES spectra of Ni-N-C and NC. **f** FT-EXAFS fitting results of Ni-N-C and NiPc. Inset shows the local coordination structure of Ni sites in Ni-N-C.

defective, which was probably caused by the asymmetric and unsaturated Ni-N coordination[29,30]. The feature II (corresponding to the dipole-allowed $1s–4p_z$ transition in $D_{4h}$ configuration) of Ni-N-C was significantly weaker than that of NiPc, also indicating the structural asymmetry of Ni sites in Ni-N-C. The intensity ratio of feature III to feature IV (corresponding to the $1s–4p_{x,y}$ transition and multiple scattering processes, respectively) represented the significant displacement of the Ni sites; its increase further reflected the existence of structural asymmetry[31]. The above analyses have confirmed the low oxidation state of Ni sites in Ni-N-C, and these Ni sites had an asymmetric and defective coordination environment. N K-edge XANES and Fourier transform extended X-ray absorption fine structure (FT-EXAFS) spectra clearly determined the coordination structure of Ni sites in Ni-N-C. As shown in Fig. 1e, the peaks corresponding to graphitic N (π*) and σ* of Ni-N-C and NC were very close, while the pyridinic N (π*) and pyrrolic N (π*) peaks of Ni-N-C were significantly weaker than that of NC. This implied that the Ni sites in Ni-N-C were mainly coordinated to pyridinic N and pyrrolic N[32]. This speculation was further supported by N 1s XPS spectra, in which a slight shift to higher energies of pyridinic and pyrrolic N peaks after Ni incorporation was observed, implying the transfer of partial electrons from N to Ni (fig. S2d). FT-EXAFS fitting results showed that the total coordination number of Ni sites in Ni-N-C was about 2.2 (Figs. 1f, S7 and S8, table S3). Notably, NiPc exhibited a signal at 2.6 Å, which was dominated by contributions from multiple scattering of ligands; for Ni-N-C, the Ni sites were distributed in the defective carbon planes, thus scattering contribution from the second coordination layer was significantly suppressed due to the disorder effects (fig. S7b)[33]. Finally, computed pre-edge spectrum was consistent with the experimental spectrum, further supporting the structure mentioned above (fig. S9). In short, combined N K-edge XANES, Ni K-edge FT-EXAFS, XPS, and computational analysis have well confirmed that the obtained Ni-N-C was composed of unsaturated Ni single sites with pyrrolic and pyridinic N double coordination. The corresponding local structure is illustrated in Fig. 1f.

## Electrochemical CO2RR performance evaluation

The performance of various catalysts for CO2RR was assessed in CO2-saturated 0.5 M potassium bicarbonate ($KHCO_3$) electrolyte using a home-made H-type electrochemical cell (fig. S10). The CO2RR activity and selectivity of Ni-N-C were evaluated through chronoamperometric testing, while NiPc and NC were analyzed comparatively (fig. S11 and S12). The main products were CO and hydrogen ($H_2$), with no appreciable liquid products (fig. S13). The total Faradaic efficiency of CO and $H_2$ ($FE_{CO}$ and $FE_{H2}$) for all catalysts were close to 100% at the measured potentials. NC exhibited unsatisfactory activity in a wide potential range. Ni-N-C showed great potential to generate CO at a low potential of −0.3 $V_{RHE}$, with the highest $FE_{CO}$ value of 98.5% achieved at −0.7 $V_{RHE}$ (Fig. 2a). NiPc exhibited only 67.9% of $FE_{CO}$ at −0.7 $V_{RHE}$ (Fig. 2a). At very negative potentials, the $FE_{CO}$ tends to decrease due to the competitive hydrogen evolution reaction (HER) and an insufficient supply of $CO_2$[34]. Beyond selectivity, Ni-N-C also demonstrated a higher partial current density of CO ($j_{CO}$) than NiPc (Fig. 2b, fig. S14). For example, Ni-N-C delivered a $j_{CO}$ of 37.6 mA $cm_{geo}^{-2}$ at −0.7 $V_{RHE}$, which was significantly higher than that of NiPc (6.6 mA $cm_{geo}^{-2}$). Ni-N-C outperformed the catalysts reported under the same electrochemical conditions (table S4). The Tafel plot analysis indicated Ni-N-C had faster electrode kinetics with a much lower slope than NiPc in the low potential region (103 mV dec$^{-1}$ versus 129 mV dec$^{-1}$, fig. S15). It is well known that the CO2RR performance of catalysts is largely limited by the slow mass transfer in H-type cell[19]; therefore, Ni-N-C was deposited on a gas diffusion electrode (GDE) for flow cell testing (fig. S16). We used 1.0 M potassium hydroxide (KOH) as electrolyte, because low concentration of protons (high pH value) is favorable to inhibit the competition of HER[35]. GDE testing showed that the $FE_{CO}$ of Ni-N-C still exceeded 90% in a wide potential range (−0.15 to −0.9 $V_{RHE}$), even reaching almost 100% at −0.6 $V_{RHE}$, indicating that Ni-N-C had good selectivity for CO2RR (Fig. 2a). The $j_{CO}$ of Ni-N-C was also significantly improved, reaching 20.1 mA $cm_{geo}^{-2}$ at −0.15 $V_{RHE}$, which was the lower potential value reported to date for achieving the same $j_{CO}$ (table S5). As the potential dropped, the $j_{CO}$ of Ni-N-C kept increasing, reaching as high as 1378.3 mA $cm_{geo}^{-2}$ at −1.0 $V_{RHE}$, which was also the higher value reported to date (Fig. 2b)[15,36,37]. The evaluated cathodic energy efficiency (CEE) of Ni-N-C in a flow cell was comparable to, or superior to, previously reported catalysts (fig. S17).

The activity of Ni-N-C and NiPc was further evaluated by TOF, assuming all Ni sites were catalytically active. From the results of H-type cell testing, the TOF values of Ni-N-C were significantly higher than that of NiPc in the applied potential range, and outperformed Ag and most reported Ni-related catalysts (Fig. 2c)[14,16,29,36,38–40]. It was worth noting that Ni-N-C could exhibit high TOF values in the very negative potential region, while the most active oxide-derived Au (OD-Au) and Fe$^{3+}$-N-C catalysts reported so far could only achieve better activity at relatively positive potentials due to their instability at more negative potentials. For example, the TOF value of Ni-N-C at −0.7 $V_{RHE}$ could reach more than 37600 site$^{-1}$ h$^{-1}$, while OD-Au and Fe$^{3+}$-N-C could only achieve TOFs up to about 1300 site$^{-1}$ h$^{-1}$ and 1000 site$^{-1}$ h$^{-1}$ at potentials above −0.5 $V_{RHE}$[16]. Based on the GDE results, the activity advantage of Ni-N-C became more obvious. Its TOF values exceeded most representative single-atom catalysts that had been reported under the same electrochemical conditions. For example, the TOF value of Ni-N-C at −0.3 $V_{RHE}$ exceeded 20000 site$^{-1}$ h$^{-1}$, which was even more than 108 times higher than that of reported Ni-N$_4$/C-NH$_2$ (185 site$^{-1}$ h$^{-1}$)[36]; at −1.0 $V_{RHE}$, the TOF value of Ni-N-C even exceeded 274000 site$^{-1}$ h$^{-1}$, showing significantly better activity than other representative catalysts (Fig. 2d)[19,36,38,40,41]. The above results demonstrated that Ni-N-C could exhibit both high selectivity and high activity during CO2-to-CO electrolysis, which can be attributed to the asymmetric and unsaturated Ni single-atomic sites. Furthermore, we evaluated the stability of Ni-N-C in both H-cell and flow cell tests. In the stability tests, we observed a sudden decline in the current density of the flow cell device (fig. S18). This drop, however, cannot be solely attributed to the structural deactivation of the catalyst, since the CO selectivity did not show significant changes. Instead, we believe that the drop may be due to the salting that is expected in an alkaline environment[35]. To address this issue, we implemented a strategy of regular electrolyte refreshment every 15 h. With this approach, Ni-N-C in 1.0 M KOH was able to maintain stability for 60 h even at a current density of ~450 mA $cm_{geo}^{-2}$ (Fig. 2e). In neutral conditions, Ni-N-C also exhibited good stability, as demonstrated by its ability to maintain high performance over extended periods of time at current densities of approximately 330 mA $cm_{geo}^{-2}$ (GDE) and 40 mA $cm_{geo}^{-2}$ (H cell). Specifically, the $FE_{CO}$ of Ni-N-C in all stability tests remains consistently high (> 96%). Post structural analysis further evidenced the stability of Ni-N-C, as revealed by the maintained single-atomic structure (fig. S19). Based on the efficient CO2RR activity of Ni-N-C, we explored its aqueous zinc (Zn)-CO2 electrochemical cell (ZCEC) performance to explore its application in electrochemical energy storage. The cell was assembled with Ni-N-C as cathode and Zn foil as anode (fig. S20). During discharge procedure, a peak power density of 1.06 mW $cm_{geo}^{-2}$ could be achieved at a current density of 5.82 mA $cm_{geo}^{-2}$ (fig. S21), indicating that Ni-N-C had efficient CO2RR performance. The assembled ZCEC meantime exhibited high durability, enabling discharge-charge cycles at 2.0 mA $cm_{geo}^{-2}$ for 40 h (fig. S22).

## Operando SR-IRAS and XAFS characterization

Systematic spectroscopic and electrochemical analyses have demonstrated the fine structure and better CO2RR performance of Ni-N-C. To gain insight into the catalytic reaction mechanism of Ni-N-C in CO2RR, operando synchrotron-radiation infrared adsorption spectroscopy

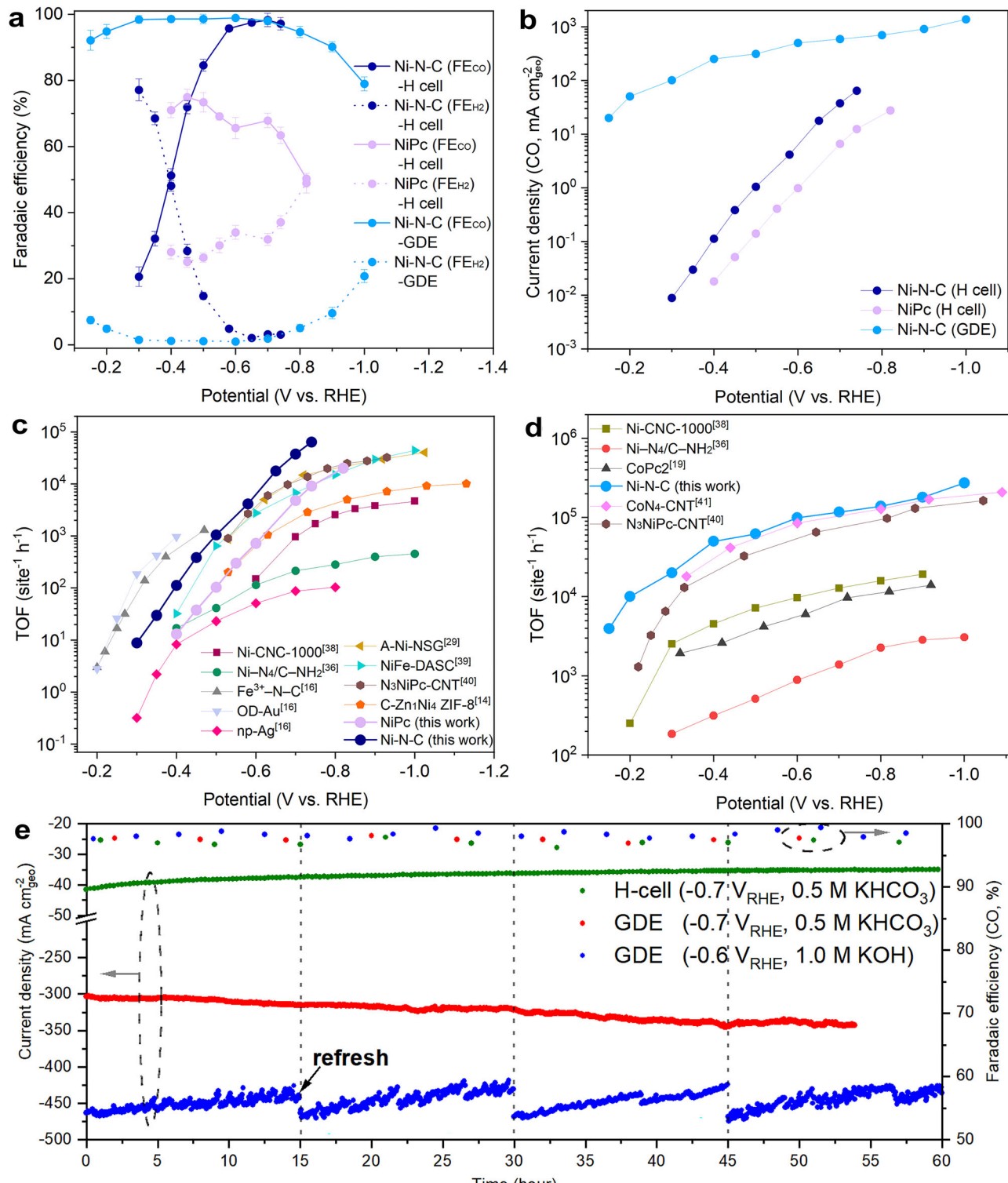

**Fig. 2 | Electrochemical CO₂RR performance. a** Measured $FE_{CO}$ and $FE_{H2}$ of Ni-N-C and NiPc. **b** Calculated $j_{CO}$ of Ni-N-C and NiPc in CO₂RR tests. Error bar in a are means ± standard deviation (3 replicates). **c**, **d** Comparison of apparent TOFs of CO generation of Ni-N-C with other reported good catalysts tested by H cell in 0.5 M KHCO₃ and by flow cell in 1.0 M KOH, respectively. **e** Chronoamperometry curves and $FE_{CO}$ of Ni-N-C measured at −0.7 $V_{RHE}$ for 60 h in H cell with 0.5 M KHCO₃ as electrolyte, −0.7 $V_{RHE}$ for over 50 h in flow cell with 0.5 M KHCO₃ as electrolyte, and −0.6 $V_{RHE}$ for 60 h in flow cell with 1.0 M KOH as electrolyte. The error bar of $FE_{CO}$ is from three independent tests.

(SR-IRAS) and XAFS tests were performed. In Fig. 3a, the SR-IRAS spectrum of Ni-N-C collected at −0.5 $V_{RHE}$ showed two adsorption peaks (**a** and **b**) located at 2075–2175 cm⁻¹, which were attributed to CO adsorption. The appearance of more than one CO adsorption peak indicated the presence of polycarbonyls. These peaks were assigned to the double adsorption of CO on Ni sites with relatively high valence (Ni^δ+, δ > 1), considering the wavenumbers of peaks **a** and **b** were slightly higher than that of Ni⁺ reported previously[42]. Therefore, the

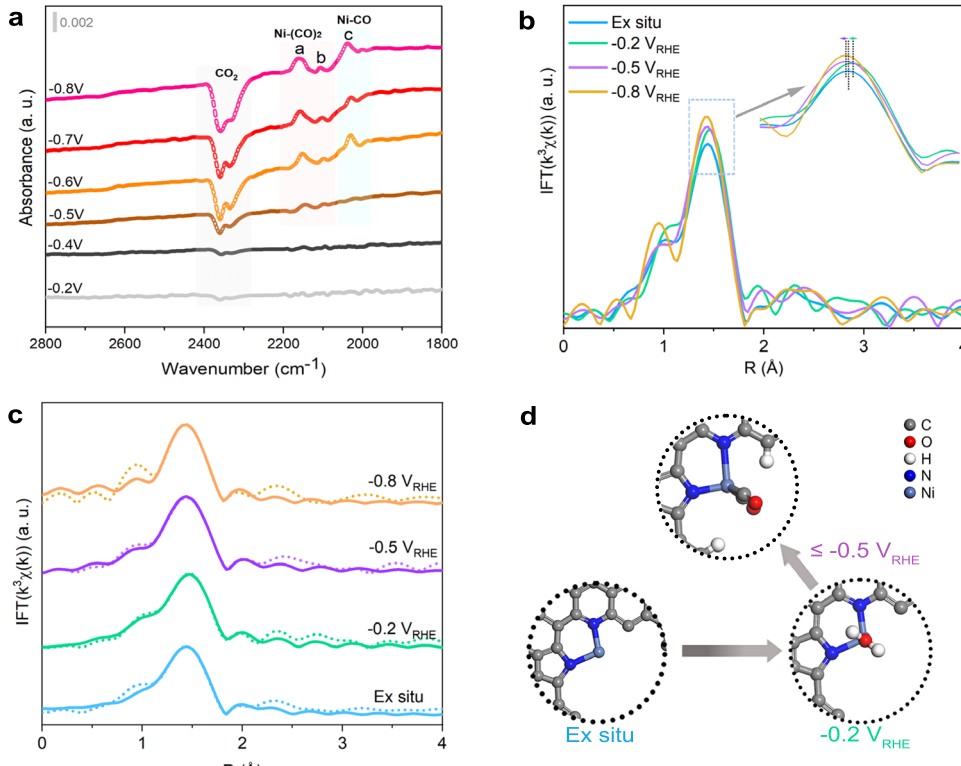

**Fig. 3 | Operando SR-IRAS and XAFS analysis under various potentials for Ni-N-C. a** SR-IRAS spectra collected among −0.2 to −0.8 $V_{RHE}$ in 0.5 M KHCO₃. **b** Ni K-edge XANES spectra at ex situ, −0.2, −0.5, −0.8 $V_{RHE}$. **c** FT-EXAFS fitting results of Ni-N-C tested at various potentials. **d** Schematic illustration of the potential-induced structure changes of low-valence Ni sites in Ni-N-C.

large cation radius allowed $Ni^{\delta+}$ sites to coordinate with more than one CO at ambient temperature (Ni-(CO)₂)[42]. When the potential was lowered to −0.8 $V_{RHE}$, the intensity of peak **a** increased significantly, while the intensity of peak **b** changed little. This consequence indicated that the properties of the two CO molecules adsorbed on Ni were different and probably caused by the electronic structural asymmetry of the Ni sites. Moreover, when the potential reached −0.6 $V_{RHE}$, another peak (**c**) appeared between 2015 and 2060 cm⁻¹, which was assigned to single CO adsorption over low-valence Ni sites ($Ni^{\Delta+}$, $\Delta < 1$). We attributed the formation of partial $Ni^{\Delta+}$ sites to the very negative voltages induced reduction of some relatively unstable Ni sites in Ni-N-C. This speculation was supported by the 1st derivative of XANES spectrum collected at −0.8 $V_{RHE}$, which showed the average valence state of Ni sites was reduced to +1.04 (fig. S23). In operando SR-IRAS results of NiPc, an observation of a broad peak located at approximately 2095 cm⁻¹ further supported the above speculation (fig. S24). However, CO adsorption on NiPc is considered to be single since four-coordinated Ni sites in NiPc possess a valence of +2, which achieves weak π-acceptor interactions with CO and results in weak CO adsorption[43,44]. Therefore, this matter is complicated and will not be extensively delved into here.

The results of ¹³CO₂ isotope experiment further confirmed the above findings. A broad peak ranging from 2255 to 2395 cm⁻¹ which corresponds to the consumption of CO₂ and ¹³CO₂ was observed (fig. S25). As the potential was reduced to −0.6 $V_{RHE}$, other peaks that correspond to CO adsorption were detected. However, due to the possibility of signal overlap between different CO adsorption peaks, we roughly assigned the detected peaks between 1960 and 2190 cm⁻¹. Our findings revealed an isotope shift of approximately 48 cm⁻¹, which is close to the reported value[45]. The ¹³CO₂ isotope experimental results are consistent with aforementioned SR-IRAS findings (Fig. 3a), which strengthens the validity of our research. Given that the double

adsorption peaks of CO (**a** and **b**) came first, while single adsorption peak of CO (**c**) came later, it is speculated that the happening of CO₂RR over $Ni^{\delta+}$ sites is easier. As for why $Ni^{\delta+}$ sites are easier for CO₂RR, it is speculated that single CO adsorption may optimize the electronic structure of $Ni^{\delta+}$ sites, thereby promoting CO₂RR kinetics on account of the observation of double CO adsorption. Operando XAFS analysis also verified the CO adsorption property of Ni-N-C, with changes in the local structure of Ni sites observed at different potentials. The analysis of XANES spectra at different potentials found that the peaks I and II slightly changed, indicating that the asymmetric and unsaturated coordination structure of Ni sites in Ni-N-C was still maintained during the reaction (fig. S26). The mildly altered intensity ratios of peaks III to IV indicated the slight shifts of Ni sites occurred. The corresponding FT-EXAFS results further demonstrated the variation of the local structure of Ni sites with potentials (Fig. 3b). In ex-situ FT-EXAFS spectrum, the main peak of the Ni-N path was located at ~1.44 Å. When the potential was −0.2 $V_{RHE}$, the position of the main peak moved to ~1.47 Å. Below −0.4 $V_{RHE}$, given that high FE$_{H2}$ and missing IR signals of CO adsorption, the Ni sites in Ni-N-C should coordinate with water molecules at −0.2 $V_{RHE}$. When the potential rose to −0.5 $V_{RHE}$, the main peak exhibited a shift to the left and an increase in intensity, which was due to the formation of shorter Ni-C bonds by CO adsorption. When the potential reached −0.8 $V_{RHE}$, the peak intensity was further enhanced, which was due to high-concentration CO accumulation. In addition, the FT-EXAFS fitting results showed that the average coordination number of Ni was increased by 0.25 at −0.5 $V_{RHE}$ and 0.36 at −0.8 $V_{RHE}$ relative to that at ex-situ, which directly confirmed the CO adsorption state of Ni sites in CO₂RR (Fig. 3c and table S6). Calculated changes of Ni-N bond lengths also show the consistent trends with FT-EXAFS fitting results (table S7). The increased coordination number also reflected the happening of increased coverage of CO at −0.8 $V_{RHE}$, consistent with these SR-IRAS results. When the potential was applied

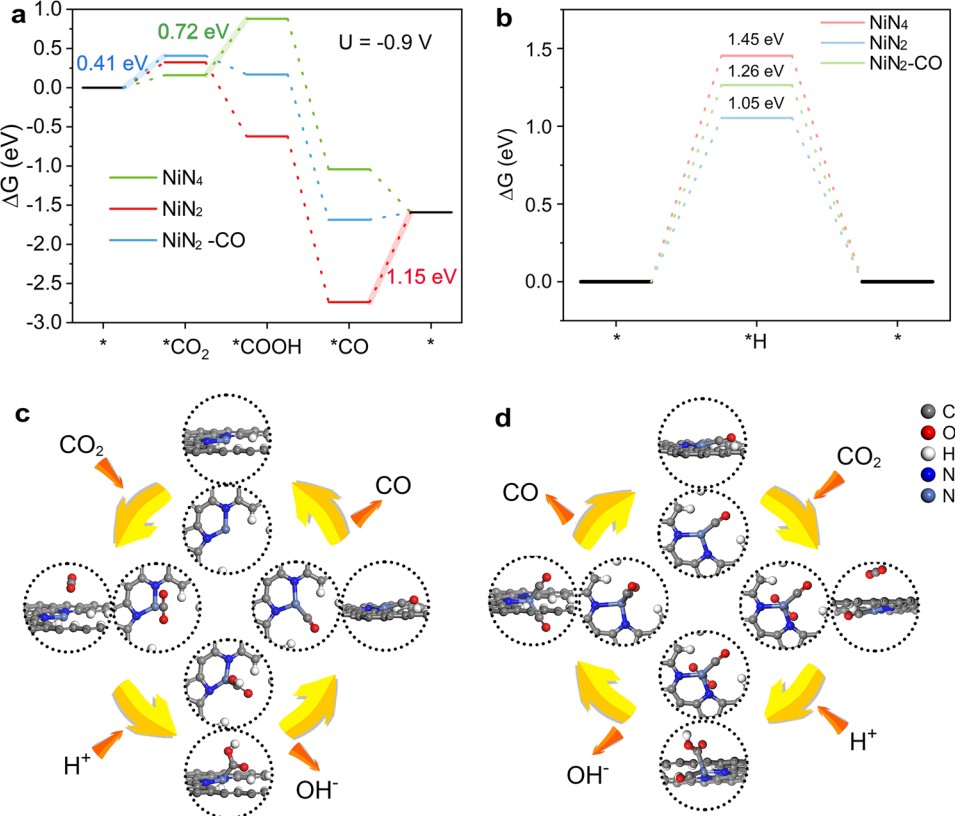

**Fig. 4 | Theoretical calculations. a** DFT calculated reaction free energy diagrams of $NiN_4$, $NiN_2$, and $NiN_2$-CO for $CO_2RR$. **b** Free energy diagrams of $NiN_4$, $NiN_2$ and $NiN_2$-CO for HER. **c**, **d** Schematic illustration of the $CO_2RR$ process over $NiN_2$ and $NiN_2$-CO sites, respectively.

back to the open circuit potential (OCP), the position of the main peak was only weakly shifted to the right and increased, which indicated that the adsorbed species at the Ni site was significantly reduced outside the operating voltage, and the main adsorbed species was CO (fig. S27). However, when operando testing was performed in Ar-saturated electrolyte, the main peak corresponding to FT-EXAFS shifted significantly to the right only at $-0.8\ V_{RHE}$, which may be caused by the adsorption of oxygen-containing species (fig. S28). It should be noted that only a portion of the catalytic sites react with reactants, which presents a challenge for accurately evaluating reaction processes using XAFS results that provide bulk average information. However, the change information obtained by operando XAFS can well support the findings from operando SR-IRAS. It should be noted that the catalyst loading used for operando tests is high, thus signal changes might not be obvious as the practical measurements of $CO_2RR$. Nonetheless, this difference should not affect the emergence and evolutionary trend of signals. Based on the above findings, we propose that the single CO adsorbed Ni sites in Ni-N-C should be the main active configuration, enabling high $CO_2$-to-CO activity. Figure 3d showed the potential-induced structural changes of Ni sites in Ni-N-C.

## Theoretical calculations

The structural properties and energy diagrams of Ni sites in $CO_2RR$ were investigated by density functional theory (DFT) calculations. According to the experimental results, $NiN_4$ and $NiN_2$ configurations were established as research objects (fig. S29). Differential charge density analysis confirmed the asymmetric electronic structure of $NiN_2$, which was different from $NiN_4$ with symmetric one (fig. S30). Bader charge analysis showed that the valence state of Ni in $NiN_2$ was lower than that in $NiN_4$ (table S8). The above analysis was in good agreement with the experimental results. In order to understand the differences of $NiN_2$ and $NiN_4$ in $CO_2RR$, the free energy diagrams of

$NiN_2$ and $NiN_4$ were calculated (Fig. 4a). For $CO_2RR$, the reaction pathway first undergoes $CO_2$ adsorption on catalytic site to form $*CO_2$, and then $*CO_2$ converts to $*COOH$ as an intermediate. Next, $*COOH$ will be broken to form $*CO$. Surely, this pathway has experienced proton and electron transfer. The free energies of the $*CO_2$-$*COOH$-$*CO$ pathway of $NiN_2$ continued to decrease, while $NiN_4$ exhibited an energy barrier of 0.72 eV during the formation of $*COOH$ from $*CO_2$, indicating that the electrochemical reaction kinetics of $NiN_2$ was superior to that of $NiN_4$. However, $NiN_2$ exhibited a huge energy barrier of 1.15 eV during CO desorption, and the difficult CO desorption process made the reactivity insufficient. This finding suggests that there is a certain reaction process with better reaction kinetics. Further analysis found that the strong CO single adsorption was beneficial to the adsorption optimization of the second CO (forming double CO adsorption state); in the meantime, the first adsorbed CO showed a favorable energetic effect on the unsaturated Ni sites. To understand this energetic effect, the projected density of states (PDOS) of Ni 3$d$ orbitals and CO molecular orbitals before and after adsorption were calculated (fig. S31). Compared with $NiN_4$, the higher $d$-band center of Ni-$N_2$ can form stronger metal-CO bond than the lower-lying Ni $d$ states of Ni-$N_4$. Then, the effect of single CO-adsorbed $NiN_2$ configuration ($NiN_2$-CO) on $CO_2RR$ was analyzed. The first CO adsorption lowers the $d$-band center of Ni, which subsequently reduces the preference for the second CO adsorption. Thus, the second CO can be released with reduced barrier (Fig. 4a). In detail, the free energies of the $*CO_2$-$*COOH$-$*CO$ pathway of $NiN_2$-CO also continued to decrease, while the the desorption of CO on $NiN_2$-CO became significantly easy compared to Ni-$N_2$. Additionally, single CO adsorption suppressed the hydrogen evolution reaction (HER) on Ni-$N_2$, which was manifested as an increase of the Gibbs free energy ($\Delta G$) of $*H$ (Fig. 4b). Furthermore, the second CO adsorption can create a larger energy barrier than CO emission, implying single CO adsorption is the best for $CO_2RR$ reaction occurs on Ni-$N_2$ sites

(fig. S32). Therefore, the obtained good $CO_2RR$ activity in this study could be attributed to the optimized kinetics of $NiN_2$ after single CO adsorption. Based on the analysis, possible pathways for $CO_2RR$ to occur on $NiN_2$ and $Ni\text{-}N_2\text{-}CO$ were proposed (Fig. 4c, d). The activity of $Ni\text{-}N_2$ sites during the $CO_2RR$ reaction is triggered by one CO adsorption.

In summary, this study has demonstrated that asymmetric and unsaturated Ni single sites could enable efficient $CO_2$ electroreduction. Structural optimization under electrochemical working conditions could play a crucial role in regulating the adsorption and activation of key intermediates on catalysts, thereby achieving good performance. Our study provides a unique insight into the self-optimization of local coordination configuration of catalysts during $CO_2RR$, which will guide the development of advanced catalysts in the future.

## Methods

### Synthesis of Ni-N-C, NiPc and NC

The Ni-N-C was obtained with the following procedure. In brief, 0.25 g glucose, 5.0 g dicyandiamide and 3.24 mg nickel chloride ($NiCl_2$) were dispersed into 100 mL deionized (DI) water to obtain a uniform solution, then the solution was heated at 80 °C for 48 h. The obtained mixture after heating was frozen with liquid nitrogen and freeze dried. Next, the dried sample was annealed at 900 °C for 2 h under argon atmosphere with a ramping rate of 5 °C/min, then the sample was heated at 700 °C for 1 h under ammonia atmosphere with a ramping rate of 5 °C/min to obtain Ni-N-C. Metal-free NC was synthesized via the similar procedure, except that no metal salt was added. To obtain NiPc, 50 mg NC and 15 mg NiPc were dissolved into 50 mL dimethyl formamide (DMF) and treated with sonication for 2 h; then the obtained suspension was stirred at room temperature for 24 h. The solid sample was collected by suction filtration with DMF, ethanol and water in sequence and freeze dried.

### Characterization

X-ray diffraction (XRD) was performed on a Philips X'pert Pro super X-ray diffractometer with Cu $K_\alpha$ Radiation ($\lambda = 1.54178$ Å). High angle annular dark field scanning transmission electron microscopy (HAADF-STEM) was performed on a 200 kV JEOL JEM-ARM200F equipped with a double spherical aberration corrector. Elemental mapping was also collected. X-ray photoelectron spectroscopy (XPS) measurements were performed on the photoelectron endstation of Hefei Light Source (HLS) with C$Is$ energy level as reference to correct binding energy. Inductively coupled plasma atomic emission spectroscopy (ICP-AES, PerkinElmer, Optima 7300 DV) measurement was used to determine the metal loading amounts. X-ray emission spectra were collected at the 4W1B beamline of Beijing Synchrotron Radiation Facility (BSRF). N K-edge X-ray absorption near edge structure (XANES) spectra were analyzed at the XMCD beamline of HLS. The Ni K-edge X-ray absorption fine structure (XAFS) spectroscopy was collected at the 1W1B beamline of BSRF.

### In situ SR-IRAS and XAFS tests

In situ SR-IRAS measurement was performed at the infrared beamline BL01B of the HLS through a homemade top-plate cell-reflection infrared set-up with a ZnSe crystal as the infrared transmission window (cut-off energy of $-625$ $cm^{-1}$). This end station was equipped with an FTIR spectrometer (Bruker 70 v/s) with a KBr beam splitter and various detectors (herein, a liquid-nitrogen-cooled mercury cadmium telluride detector was used) coupled with an infrared microscope (Bruker Hyperion 2000) with an ×15 objective. The catalyst electrode was tightly pressed against the ZnSe crystal window with a micrometre-scale gap to reduce the loss of infrared light. To ensure the quality of spectra, the apparatus adopted a reflection mode with a vertical incidence of infrared light. Each infrared absorption spectrum was acquired by averaging 128 scans at a resolution of 4 $cm^{-1}$. The

background spectrum of the catalyst electrode was acquired at an open-circuit voltage before each systemic measurement, and the measured potential range was $-0.2$ to $-0.8$ $V_{RHE}$. All the electrochemical tests were performed in $CO_2$-saturated 0.5 M $KHCO_3$ electrolyte. Isotope experiments were carried out in $^{13}CO_2$-purged 0.5 M $KHCO_3$, and the measured potential range was $-0.4$ to $-1.0$ $V_{RHE}$. The sample loading amount of each test was 40 μg (20 μL).

In situ fluorescence-mode XAFS was performed at 1W1B beamline of BSRF. All the X-ray was monochromatized by a double crystal Si (111) monochromator. The energy of the Ni K-edge spectra was calibrated by Ni foil. The catalyst modified carbon paper was used as the working electrode and was mounted on a home-made electrochemical cell. A peristaltic pump was used to maintain the flow of Ar or $CO_2$-saturated 0.5 M $KHCO_3$ during the test to ensure mass transfer. The obtained XAFS data were processed and analyzed according to the standard procedures by using the WinXAS 3.1 program[46]. Theoretical amplitudes and phase-shift functions were calculated with the FEFF8.2[47]. The fitting results were obtained by setting different structure parameters until that the fitting data coincide well with experimental data. The sample loading amount on carbon paper was about 1.0 mg $cm_{geo}^{-2}$, and the potentials were set as $-0.2$ $V_{RHE}$, $-0.5$ $V_{RHE}$, $-0.8$ $V_{RHE}$ and OCP.

### Electrochemical measurements

Electrochemical measurements were carried out on CHI series potentiostat. All potentials in this study were calibrated by a hydrogen electrode (PHY-RHE). The $CO_2$ electroreduction experiments were carried out in an H-cell separated by Nafion 117. Carbon paper with a loaded sample (0.5 $cm^2$) was used as the working electrode. The brief procedure of electrode preparation was as below: 2 mg of the sample was dispersed in the mixed solution of DI water (0.55 mL), isopropyl alcohol (0.25 mL) and 0.5 wt% Nafion solution (0.2 mL), followed by the sonication for 30 min to obtain a homogeneous ink. A measure of 25 μL of the ink was drop-casted onto carbon paper (loading density = 0.1 mg $cm_{geo}^{-2}$), and dried at room temperature. The Ag/AgCl and Ni foam were used as the reference and counter electrodes. The electrolyte was $CO_2$-saturated 0.5 M $KHCO_3$. During the test, $CO_2$ was continuously fed into the cathode chamber and the magnetic stirrer was operated at 1000 rpm to accelerate mass transfer. 80% resistance compensation was applied in our measurements.

Before other measurements, the working electrodes executed cyclic voltammetry of 50 cycles with a scan rate of 0.1 V $s^{-1}$ within the potential range of $+0.4$ V ~ $-0.3$ $V_{RHE}$ to reach a steady state. Then, a CV curve within the potentials range of 0 V ~ $-0.8$ $V_{RHE}$ was collected with a scan rate of 50 mV $s^{-1}$. The chronoamperometry (CA) was used to study the current response with different potentials, and the current value within 30 min was regarded as the steady state to do further activity and selectivity analysis.

Flow cell measurements were performed in a commercial flow cell reactor and consisted of 4 parts. Gas diffusion electrode (GDE) with loaded sample (0.5 mg $cm_{geo}^{-2}$) was used as the cathode, Ag/AgCl (saturated KCl) electrode equipped with a salt bridge and Ni foam (4 cm * 2 cm) were used as the reference and counter electrodes, respectively. The anolyte compartment and the catholyte compartment is separated by an anion exchange membrane. During the test, the $CO_2$ feed gas was introduced into the gas chamber with a constant flow rate of 20 sccm, and the gas outlet was connected to the GC to quantify gas products. The electrolyte in the cathode and anode chambers was circulated by using a gas-liquid mixed flow pump.

The gas products were analyzed by an online gas chromatography (GC, Agilent 7890B) with thermal conductivity detector (TCD) and flame ionization detector (FID). Liquid electrolytes were analyzed by $^1H$ nuclear magnetic resonance (NMR) using dimethylsulphoxide (DMSO) as an internal standard after $CO_2$ reduction electrolysis for 30 min. The Faradaic efficiency (FE) of CO or $H_2$ was calculated using the following

equation:

$$FE_i = 2\frac{p_0 \nu x_i F}{IRT} \qquad (1)$$

where, $p_o$ is 101 kPa; $\nu$ is the gas flow rate; $x_i$ is the fraction of gas detected by GC; $F$ is the Faraday constant (96485.3 C mol$^{-1}$); $I$ is the stable current; $T$ is 273 K; $R$ is the gas constant (8.314 J mol$^{-1}$ K$^{-1}$).

The apparent turnover frequency (TOF) of CO formation was calculated according to:

$$TOF(h^{-1}) = \frac{I_{CO}/2F}{m_{cat} \times w/M_{Ni}} \times 3600 \qquad (2)$$

where, $I_{CO}$ is partial current for CO production; $F$ is the Faraday constant (96485.3 C mol$^{-1}$); $m_{cat}$ is the mass of catalyst on the electrode; $w$ is the mass fraction of Ni in catalyst; $M_{Ni}$ is the atomic mass of Ni (58.69 g mol$^{-1}$). In this study, $m_{cat}$ is 50 μg for H-cell test, and is 250 μg for GDE test.

The cathodic energy efficiency (CEE) of CO is calculated using the following equation:

$$CEE_{CO} = \frac{1.23V - E^0_{CO}}{1.23V - E_w} \times FE_{CO} \qquad (3)$$

where, $E^0_{CO}$ is thermodynamic potential of CO; $E_w$ is working potential; $FE_{CO}$ is the faradaic efficiency of CO.

Rechargeable Zn-CO$_2$ battery tests were carried out in a home-made dual electrolyte system separated by a bipolar membrane. Catholyte was 0.8 M KHCO$_3$ saturated with CO$_2$, and anolyte was 0.8 M KOH with 0.02 M Zn(CH$_3$COO)$_2$. Catalyst-modified carbon paper was used as the cathode (0.5 mg cm$_{geo}^{-2}$), and the anode was a zinc plate (diameter was 2.0 cm). The galvanostatic discharge-charge cycling curves were measured at discharge current density of 2.0 mA cm$_{geo}^{-2}$ and charge current density of 2.0 mA cm$_{geo}^{-2}$. In all tests, CO$_2$ was introduced into the cathode chamber with a rate of 10 sccm.

## DFT calculations
The spin-polarized density functional theory (DFT) calculations were carried out using the Vienna ab initio simulation package (VASP)[48–50]. The project augmented wave method (PAW) and Perdue-Burke-Ernzerhof (PBE) functional were employed to calculate the exchange and correlation energies[51,52]. The DFT + U method was applied for the d-electrons of Ni (U-J = 6.4 eV)[53]. The orthorhombic monolayer graphene structure was constructed by rotating the lattice vector. NiN$_2$ and NiN$_4$ were modeled using a 3 × 3 orthorhombic monolayer graphene. To simulate NiN$_2$ in edges or defects in graphene, 15 C atoms were removed. A vacuum layer of 20 Å was added to avoid periodic interference. For all models, the 2 × 2 × 1 Γ-k-point mesh was employed. The plane-wave cutoff energy and the convergence criterion of force and energy were set to 500 eV, 0.02 eV Å$^{-1}$ and 10$^{-6}$ eV. The DFT-D3 method was applied to modify the van der Waals interaction[54,55].

## Data availability
The main data supporting the findings of this study are available within the article and its Supplementary Information or are available from the corresponding authors upon reasonable request. Source data for the following figures are provided with this paper. Figures 1c–f, 2a–e, 3a–c, 4a, b. Source data are provided with this paper.

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

## Acknowledgements

S.L. thanks the National Key R&D Program of China for project 2020YFA0405803, and the National Natural Science Foundation of China for projects 12225508 and U1932201. H.Q. thanks the National Natural Science Foundation of China for project 22208336, Fundamental Research Funds for the Central Universities for project WK2310000111, and the Natural Science Foundation of Anhui Province for project 2108085QA31. L.H. thanks Fundamental Research Funds for the Central Universities for project WK2310000099. We thank Shanghai Synchrotron Radiation Facility (14W1 and 14B1, SSRF), Beijing Synchrotron Radiation Facility (1W1B, 4W1B and 4B9A, BSRF), Hefei Synchrotron Radiation Facility (Infrared spectroscopy and microspectroscopy, MCD-A and MCD-B Soochow Beamline for Energy Materials, Photoemission and Catalysis/Surface Science Endstations at NSRL) for helps in characterization.

## Author contributions

Q.H. and L.S. conceived the experiments and wrote the paper. Q.H., Y.Z.Z., H.J.L., W.J.X., Z.X.W., S.C.Q., H.H.D., D.L.C., J.F.Z., and Z.M.Q. prepared samples and performed characterization. Q.Z. and X.J.W. performed simulations of the samples. All authors discussed the results and commented on this manuscript.

## Competing interests

The authors declare no competing interests.
