## [Peer Review File · Nature Communications]

REVIEWER COMMENTS

Reviewer #1 (Remarks to the Author):

The article entitled “Asymmetric dinitrogen-coordinated nickel single-atomic sites for efficient CO₂ reduction” by Zhou et al. reports a Ni single atom catalyst which possess an asymmetric dinitrogen coordinating the metal center. The proposed catalyst exhibits extraordinary current densities and TOF numbers for the reduction of CO₂ to CO. A selection of techniques such as XRD, HAADF-STEM and elemental mapping reveal the homogeneity presence of isolated atoms. Further X-ray absorption spectroscopy analysis proposed the asymmetric character of the dinitrogen binding. However, these results are not conclusive and other techniques can be proposed to fully address the structure of the catalysts. Electrochemical results can be highly controversial, and they will require a more detailed analysis. In general, the text requires revision and more clarity in the results, discussion and conclusions. Article can be further reviewed after major corrections and revisions. Here some points to consider:

Introduction

There is a lack of references while it requires restructuring. Some typos are found which should be corrected.

Characterization

- XPS can be used to give further insights on the asymmetric nitrogen character, since it is a standard technique that allows the differentiation of N-functional groups. Therefore, it is suggested the study by XPS and inclusion of the results in the paper.

- There is a lack of explanation regarding to the determination of the average valence by linear combination fittings by XANES. Please provide more details on this point. In addition, a more quantitative way to report oxidation state changes by XANES relies on the comparison of the position of the rising edge by the inflection point of the 2nd derivative. It is suggested to include a table and the data regarding this parameter.

-Structural changes conclusions by XANES should be revised since they are speculative. Computational methods, such as pre-edge analysis, are proposed to give more solidity to the results. This point is important since previous reports by Yang et al. Nature Energy. 2018. 3. 140-147 report similar XANES spectra.

- Figure S4.B: there is no signal at around 2.6 Å (expected Ni-C) which is presented in the NiPc data. Could the authors suggest an explanation to this?

- Further information is required to complete the methods: sample collection and preparation, parameters, etc.

Electrochemistry in a H-cell

- It is suggested to present the electrochemical data in the IUPAC conversion.
- Please include cyclic voltammograms in the supplementary information to complement electrochemical data.
- Current density values are comparable with previously reported (Yang et al. Nature Energy. 2018. 3. 140-147) of approximately 40 mAcm⁻² at -0.7 V vs RHE with a similar FE(CO) 97% (current report 37.6 mAcm⁻² at -0.7 V vs RHE).
- Labelling experiments are required, as well as for in situ SR-IRAS measurements.
- It is suggested to provide characterization results after electrolysis to confirm the integrity of the catalyst.

Electrochemistry in GDE

- Current density is reported as 1378.3 mAcm⁻² at -1 V but at this potential, the FE(CO) drops. What is the integrity of the catalyst at that potential and under these conditions?
- Could the authors explain the variation of the electrolyte from KHCO₃ to KOH?
- Could the authors provide more details regarding to the long-term electrolysis study?
- How was the TOF calculated? Could the authors provide the mathematical determination of the value? Please check: Yang et al. Nature Energy. 2018. 3. 140-147 – Methods.

In situ-operando XAS

- Could the authors suggest further explanations to the almost no differences observed during reaction? If CO coordination occurs (as suggested by SR-IRAS measurements), variations in the XANES spectra (figure S15) should be noticeable, more probable at peak B (1s-4pz transition). Same if H₂O coordination is proposed at -0.2 V, differences in the spectra should be expected. Please suggest explanation to these points.
- Data under inert atmosphere is suggested to be added in order to provide comparison. In addition, data after electrolysis should be incorporated (coming back to OCP for instance).
- Suggested to include a zoom insert in figure S15 to clearly observe the variations discussed on text.

References

References are reported in different manner (some cases as et al, other cases reporting all the authors), please refer to all in the same way.

Reviewer #2 (Remarks to the Author):

In this paper, the asymmetrically coordinated Ni single atom catalyst achieved higher turnover frequency than noble metal nanoparticles. It is noticeable in the CO₂RR mechanism that a well-known poisoning ligand, CO, functions as a promoter for releasing a CO₂RR product in a thermodynamically preferable way. Overall, this work will lead the interest of the related research fields after making a few revisions.

Q1. (i) According to the EXAFS description in page 6, line 177-182, and (ii) based on the suggested CO₂RR mechanism, the number of coordinated CO molecules should increase from -0.5 V_{RHE} to -0.8 V_{RHE}, while the opposite in Fig. 3E. I think the insets for -0.5 V_{RHE} and -0.8 V_{RHE} are changed with each other.

Q2. In Fig. 4, the authors compare the thermodynamics of CO₂RR with and without the second CO molecule, according to the mechanism as *CO₂ → *COOH → *CO → * + CO.

What about the effect of the second CO on the initial adsorption of CO₂ molecule? As starting from bare (*) in the HER thermodynamics of Fig. 4B, the CO₂RR in Fig. 4A should be also drawn for a full catalytic cycle.

Q3. It is not necessary to include, but the expression in the line 208-209 can be improved by d-band center theory:

"Compared with NiN₄, Ni-N₂ displayed stronger interaction with molecular orbitals of CO, which obviously altered its electronic structure."

It can be understood by the higher d-band center of Ni²⁺ that can form stronger metal-CO bond than the lower-lying Ni d states of Ni-N₄.

Q4. (minor) Fig. S17 and S18 should be improved for readers (e.g. resolution, etc.) and there are typos (aborb, unaborb → adsorb, unadsorb). Please put the (+) signs on the Bader charges in Table S6 for clarity.

Reviewer #3 (Remarks to the Author):

Reviewer report for NCOMMS-22-49056-T

Asymmetric Dinitrogen-Coordinated Nickel Single-Atomic Sites for Efficient CO₂ Electroreduction

By Yuzhu Zhou et al.

This paper describes the synthesis of single atomic Ni sites with dinitrogen coordination, obtained by thermal treatment of graphitic carbon nitride under Ar and NH₃ environment. The catalyst is then used as electrocatalyst for CO₂ reduction (CO₂RR), placed on carbon paper and tested with H cell and flow cell configuration. A suite of operando techniques is then applied to understand the behaviour of the catalyst during CO₂RR. While the performance appear to be appealing, some improvement to the characterisation techniques, processing and interpretation is needed.

Queries:

- g-C₃N₄ is notoriously insulating, and it is only widely used for photocatalysis due to photoactivity/conductivity (doi:10.1002/anie.201403375). Could the authors explain if the state of treated g-C₃N₄ still remains or pretty much carburised? I think it is worth checking the structural and electrical properties of the g-C₃N₄ and treated NC with and without the metal to see if there is a difference in the N-doped carbon support.
- Figure 1D: and the discussion at page 3: the Ni-N-C EXAFS is rather featureless, which is expected of a highly disordered/defective local structure. It is suggested to change the naming of the EXAFS features because it can be confused with the sub-panel figure 1 numbers. Also, instead of peaks, probably “features” are more appropriate. For feature “A”, the proposed non-centrosymmetry of the Ni scatterer will cause 3d-4p mixing, which gives rise to the increased pre-edge feature “A” intensity (see e.g. doi:10.1021/bi900087w). The assessment that the Ni oxidation state is <2+ is supported.
- Performance-wise, the catalyst appears to be excellent at this scale. I’m wondering what the standing of the catalyst is if Au and Ag catalysts are also included in the comparison, as the authors claim in page 4 “Ni-N-C outperformed most reported catalyst under same electrochemical conditions”. I’m wondering if energy-efficiency calculation, especially for the flow cell (as it is done with similar conditions with others), have been performed. I think Ag is relevant to compare as it is the incumbent catalyst that has shown large current deployment and much longer stability (see e.g. doi:10.1038/s41565-020-00823-x)

- Although some degree of catalytic activity stability is displayed in Figure 2E, a pertinent question to single-atom type catalyst is the inevitable surface reconstruction and agglomeration. With the high current/turnover, I think catalyst migration and agglomeration may be unavoidable. Could the authors comment on this aspect? There are some reports suggesting various operando techniques to track, (for example doi:10.1021/acs.nanolett.0c03475; doi:10.1021/acs.chemrev.2c00495; doi:10.1038/s41929-018-0182-6), I'm wondering if such approach has been attempted (for example, concurrent with the operando XAS shown in Figure 3).

- One follow-up question for the stability: why the flow cell stability is only limited to 18 hours?

- The assignment and interpretation of SR-IRAS peaks are not as unambiguous as expected. Ni-CO (monocarbonyl) is expected around 2118 cm⁻¹ in cited ref 26 but is ascribed to 2030 cm⁻¹ in the manuscript. Perhaps, a comparison with ¹³CO isotope (and examination of peaks at slightly wider wavenumber range, say 1800 to 2800 cm⁻¹) may reveal more information and give a bit more certainty. Is it possible that the authors are looking at co-adsorption of CO and OH or H? (See e.g. DOI: 10.1016/0039-6028(94)00744-6 and DOI: 10.1016/S0167-2991(96)80058-9). I am also curious if any in-situ measurement for the alternative Ni-PC has been done, and if different behaviour of CO adsorption (or co-adsorption) is also seen on Ni-PC.

- XAS, Page 6, "The analysis of XANES spectra at different potentials found that the peaks A and B slightly changed, indicating..." This labelling of "Peaks A and B" are very confusing and is easily mistaken for the earlier IR spectroscopy. The terminology is also confusing, because actually Figure 3C is the R space derivative obtained from data in Figure S15A, which qualifies as EXAFS for the energy range is sufficiently large (8320 to 8420) to cover the oscillations (all Ni XAS shown in this work has the same energy range). The interpretation of the EXAFS data also can be improved. For example, what is the significance of ca. 0.03 Å of first neighbour (Ni-N) radial distance change? By the way the correct atomic distances are the ones listed in Table S5. What are the error values for the fitting, and if 0.03 Å is within the error? How much elongation should we be expecting in Ni-N when OH or CO is adsorbed on Ni (e.g. can this be corroborated with the calculation)?

REVIEWER COMMENTS

Reviewer #1 (Remarks to the Author):

The article entitled "Asymmetric dinitrogen-coordinated nickel single-atomic sites for efficient CO₂ reduction" by Zhou et al. reports a Ni single atom catalyst which possess an asymmetric dinitrogen coordinating the metal center. The proposed catalyst exhibits extraordinary current densities and TOF numbers for the reduction of CO₂ to CO. A selection of techniques such as XRD, HAADF-STEM and elemental mapping reveal the homogeneity presence of isolated atoms. Further X-ray absorption spectroscopy analysis proposed the asymmetric character of the dinitrogen binding. However, these results are not conclusive and other techniques can be proposed to fully address the structure of the catalysts. Electrochemical results can be highly controversial, and they will require a more detailed analysis. In general, the text requires revision and more clarity in the results, discussion and conclusions. Article can be further reviewed after major corrections and revisions. Here some points to consider:

1.1 Introduction

There is a lack of references while it requires restructuring. Some typos are found which should be corrected.

Response: Thanks. We have re-arranged the Introduction and corrected the typos in the revised manuscript.

1.2 Characterization

- XPS can be used to give further insights on the asymmetric nitrogen character, since it is a standard technique that allows the differentiation of N-functional groups. Therefore, it is suggested the study by XPS and inclusion of the results in the paper.

Response: Thanks. We have provided the N 1s XPS spectra of both Ni-N-C and NC (Fig. R1). It can be observed that the pyridinic and pyrrolic N peaks of Ni-N-C show a slight shift to higher energies, implying the transfer of partial electrons from N to Ni. These findings demonstrate the coordination of Ni with pyridinic and pyrrolic N. We have discussed the implications of these results in the revised manuscript.

Fig. R1. N 1s XPS spectra of Ni-N-C and NC

1.3 There is a lack of explanation regarding to the determination of the average valence by linear combination fittings by XANES. Please provide more details on this point. In addition, a more quantitative way to report oxidation state changes by XANES relies on the comparison of the position of the rising edge by the inflection point of the 2nd derivative. It is suggested to include a table and the data regarding this parameter.

Response: Thanks. We have provided a thorough explanation and detailed account of our valence state analysis from LCF by XANES in the revised manuscript. To further investigate the valence state of Ni, we have included 1st derivative data. However, we note that the 2nd derivative data is insufficient for calculating the valence state of Ni in our work. This is

attributed to the low loading of Ni in Ni-N-C, which negatively impacts the quality of the collected XAFS data. Nonetheless, the 1st and 2nd derivatives are equal for the valence state analysis (*Nat. Commun.* **11**, 3525 (2020)). To enhance our analysis, we integrate a Table and 1st derivative figure in the revised manuscript, accompanied by corresponding discussions.

Fig. R2. 1st derivative data of Ni K-edge XANES for Ni foil, Ni-N-C, and NiPc.

Table R1. The summary of edge energies and corresponding valence states from 1st derivative data of Ni K-edge XANES.

Sample	Ni foil	Ni-N-C	NiPc
Edge energy (eV)	8333.00	8337.30	8339.28
Valence states	0.00	+1.37	+2.00

1.4 Structural changes conclusions by XANES should be revised since they are speculative. Computational methods, such as pre-edge analysis, are proposed to give more solidity to the results. This point is important since previous reports by Yang et al. *Nature Energy*. 2018. 3. 140-147 report similar XANES spectra.

Response: Thanks. We have revised the discussions about XANES. In our work, we provide a detailed analysis of the local structure surrounding Ni in Ni-N-C, providing evidence for the two-coordination structure of Ni sites. Firstly, complementary Ni K_{β} XES and Ni K-edge XANES analyses suggest the presence of low-valence Ni sites via the positive energy shift in Ni K_{β} XES/N 1s XPS spectra and low absorption energy in Ni K-edge XANES (Fig. R1, R3A,B). Secondly, FT-EXAFS fitting reveals a coordination number of approximately 2.2 for Ni in Ni-N-C (NiPc with Ni-N₄), while N K-edge XANES suggests the maintenance of graphitic N and a decrease in pyrrolic and pyridinic N after Ni incorporation (Fig. R3C,D). As a supplement, pre-edge simulation is carried out. The simulated XANES spectrum of Ni for Ni-N-C is highly consistent with the experimental data, demonstrating the predominant moiety of Ni atom in the optimized catalyst (Fig. R4). These results solidly demonstrate the asymmetric N coordination of Ni in Ni-N-C, supporting the proposed structure as the most likely. Furthermore, our sample is distinct from the Ni single-atom catalysts reported by Yang et al. To facilitate comparison, we include their spectra below, which reveal clear differences between the XAS spectra of the samples (Fig. R5).

Fig. R3. (A) XES spectra of Ni-N-C, NiPc, and NiO. Inset is the zoom in of $K_{\beta 2}$ located at the yellow region. (B) Ni K-edge XANES spectra of Ni-N-C, Ni foil, and NiPc. (C) N K-edge XANES spectra of Ni-N-C and NC. (D) The FT-EXAFS fitting results of Ni-N-C and NiPc.

Fig. R4. Comparison between the experimental XANES spectrum of Ni and the simulated spectrum.

Fig. R5. XANES and FT-EXAFS spectra of A-Ni-NG, A-Ni-NSG, and Ni-N-C.

1.5 Figure S4.B: there is no signal at around 2.6 Å (expected Ni-C) which is presented in the NiPc data. Could the authors suggest an explanation to this?

Response: Thanks. FT-EXAFS is a technique that results from the modulation of photoelectrons by surrounding atoms at various distances around the element being analyzed. The contributions of long-range scattering paths is typically more significant than those of short-range scattering paths (*Chem. Rev.* **121**, 882-961 (2020)). In the case of NiPc with a well-defined structure, the signal at 2.6 Å arises mainly from the multiple scattering of ligands. In the case of Ni-N-C, the Ni sites are dispersed among the defective carbon planes. The scattered contribution from the second coordination layer is considerably suppressed because of the effects of disorder. This issue has been further elaborated in the revised manuscript.

1.6 Further information is required to complete the methods: sample collection and preparation, parameters, etc.

Response: Thanks. We have supplemented the methods with further experimental details in the revised manuscript.

1.7 Electrochemistry in a H-cell

- It is suggested to present the electrochemical data in the IUPAC conversion.

Response: Thanks. We have revised the data in the revised manuscript with reference to the IUPAC recommendation. A representative legend is shown below (Fig. R6).

Fig. R6. (A) The measured FE_{CO} and FE_{H2} of Ni-N-C and NiPc, and (B) the calculated j_{CO} of Ni-N-C and NiPc in CO₂RR tests. Comparison of apparent TOFs of CO generation of Ni-N-C with other reported good catalysts tested by (C) H cell in 0.5 M KHCO₃ and by (D) flow cell in 1.0 M KOH. (E) Chronoamperometry curves and FE_{CO} of Ni-N-C measured at -0.7 V_{RHE} for 60 hours in H cell with 0.5 M KHCO₃ as electrolyte, -0.7 V_{RHE} for over 50 hours in flow cell with 0.5 M KHCO₃ as electrolyte, and -0.6 V_{RHE} for 60 hours in flow cell with 1.0 M KOH as

electrolyte. The error bar of FE_{CO} is from three independent tests.

1.8 Please include cyclic voltammograms in the supplementary information to complement electrochemical data.

Response: Thanks. We have included CV data in the Supplementary Materials.

Fig. R7. CV curves of Ni-N-C, NiPc, and NC in Ar and CO_2 saturated 0.5 M $KHCO_3$ at a scan rate of 50 mV s^{-1} .

1.9 Current density values are comparable with previously reported (Yang et al. Nature Energy. 2018. 3. 140-147) of approximately 40 mA cm^{-2} at -0.7 V vs RHE with a similar $FE(CO)$ 97% (current report 37.6 mA cm^{-2} at -0.7 V vs RHE).

Response: Thanks. One of the best levels of performance for Ni-based single-atom catalysts in CO_2 -to- CO conversion has been achieved by Yang et al. However, the current density used in their study is normalized based on the geometric area of the electrode, which is affected by the interfacial active site density (metal loading) and the loading amount of catalyst under actual testing conditions (Supplementary Figure 6, Tables 1 in Nat. Energy 3, 140-147 (2018)). To better confirm the intrinsic activity difference between different catalytic sites, the TOF values can be calculated. We have calculated the TOF values at -0.7 V_{RHE} for both catalysts using the same procedure (Table R2).

Table R2. Comparison of apparent TOFs of CO generation of Ni-N-C with A-Ni-NSG.

Sample	Ni-N-C	A-Ni-NSG
TOF at -0.7 V_{RHE} ($\text{site}^{-1} \text{ h}^{-1}$)	~ 37300	~ 11900

Our results indicate that the Ni sites in Ni-N-C possess better intrinsic activity. In Fig. 2C of main text, the TOF comparison between our catalyst and Yang et al.'s catalyst (A-Ni-NSG) is exhibited. Significantly, our catalyst displays a much higher activity than A-Ni-NSG.

1.10 Labelling experiments are required, as well as for in situ SR-IRAS measurements.

Response: Thanks. We have performed $^{13}CO_2$ isotope experiment. Based on the experiment with $^{13}CO_2$ isotope, we have observed a broad peak ranging from $2255\text{-}2395 \text{ cm}^{-1}$ which corresponds to the consumption of CO_2 and $^{13}CO_2$ (Fig. R8). As the potential was reduced to -0.6 V_{RHE} , we also detected other peaks that correspond to CO adsorption. However, due to the possibility of signal overlap between different CO adsorption peaks, we roughly assigned the detected peaks between $1960\text{-}2190 \text{ cm}^{-1}$. Our findings revealed an isotope shift of approximately 48 cm^{-1} , which is close to the reported value (*ACS Energy Lett.* **4**, 682-689 (2019)). Importantly, these results are consistent with our previous SR-IRAS findings, which strengthens the validity of our research. Isotope data and our discussion on this matter has been included in the revised manuscript.

Fig. R8. In situ isotope analysis for Ni-N-C in $^{13}\text{CO}_2$ -purged 0.5M KHCO_3 .

1.11 It is suggested to provide characterization results after electrolysis to confirm the integrity of the catalyst.

Response: Thanks. We have provided characterization results of Ni-N-C after operation at $-1.0 V_{\text{RHE}}$ for 10 h. XRD and HAADF-STEM analysis show that Ni-N-C maintains atomic level dispersion within the desired potential range (Fig. R9). Our discussion on this matter has also been included in the revised manuscript.

Fig. R9. (A) XRD and (B) HAADF-STEM analysis of Ni-N-C after operation.

1.12 Electrochemistry in GDE

- Current density is reported as $1378.3 \text{ mA cm}^{-2}$ at -1 V but at this potential, the $\text{FE}(\text{CO})$ drops. What is the integrity of the catalyst at that potential and under these conditions?

Response: Thanks. The decrease in CO_2 selectivity observed at high potentials can be attributed to the strong competition with hydrogen evolution reaction (HER) at large overpotentials. Furthermore, the supply of CO_2 may be insufficient at these high reaction rates, which can be explained by an equilibrium between the thermodynamics of the electrode potential and the diffusive dynamics of the feed gas. Similar observations have been reported in other studies (*Nat. Commun.* **10**, 3602 (2019); *Nat. Catal.* **2**, 1124-1131(2019)).

The catalyst after operation is subjected to XRD and HAADF-STEM analysis, which confirm that the single-atomic structure of Ni-N-C is maintained under these conditions (Fig. R9). This suggests that the major CO_2RR activity stems from single Ni sites.

1.13 Could the authors explain the variation of the electrolyte from KHCO_3 to KOH ?

Response: Thanks. The inhibition of the competition of HER is facilitated by the low concentration of protons, corresponding to a high pH value. As a result, we conducted GDE testing in a KOH aqueous solution (*ACS Energy Lett.* **3**, 2527-2532 (2018) ; *JACS* **143**, 3245-3255 (2021)). Further details have been included in the revised manuscript.

1.14 Could the authors provide more details regarding to the long-term electrolysis study?

Response: Thanks. To provide a more comprehensive understanding of our experimental procedure, we provide a detailed description of the testing process. Additionally, we have

included the complete stability curves in the revised manuscript to supplement our results. During our stability testing in KOH, we observed a sudden decline in the current density of the flow cell device. This drop, however, cannot be solely attributed to the structural deactivation of the catalyst, since the CO selectivity did not show significant changes (Fig. R10A). Instead, we believe that the drop may be due to the salting that is expected in an alkaline environment (*JACS* **143**, 3245-3255 (2021)). To address this issue, we implemented a strategy of regular electrolyte refreshment every 15 hours. With this approach, Ni-N-C in 1.0 M KOH was able to maintain stable for 60 hours even at a current density of $\sim 450 \text{ mA cm}_{\text{geo}}^{-2}$ (Fig. R10B). In neutral conditions, Ni-N-C also exhibited good stability, as demonstrated by its ability to maintain high performance over extended periods of time at current densities of approximately $330 \text{ mA cm}_{\text{geo}}^{-2}$ (GDE) and $40 \text{ mA cm}_{\text{geo}}^{-2}$ (H cell). Specifically, the FE_{CO} of Ni-N-C in all stability tests remains consistently high ($>96\%$).

Fig. R10. (A) Chronoamperometry curve and FE_{CO} of Ni-N-C measured at $-0.5 V_{\text{RHE}}$ for over 30 hours in GDE with 1.0 M KOH as electrolyte. (B) Chronoamperometry curves and FE_{CO} of Ni-N-C measured at $-0.7 V_{\text{RHE}}$ for over 50 hours in both H cell and GDE with 0.5 M KHCO_3 as electrolyte, and $-0.6 V_{\text{RHE}}$ for 60 hours in GDE with 1.0 M KOH as electrolyte. Refresh means the refreshment of electrolyte at that time.

1.15 How was the TOF calculated? Could the authors provide the mathematical determination of the value? Please check: Yang et al. *Nature Energy*. 2018. 3. 140-147 – Methods.

Response: Thanks. We have added the details of the TOF calculation in the Note at the end of Supplementary Materials.

To calculate the TOF of CO formation in literature, we used the formula:

$$\text{TOF}_{\text{CO}} = I_{\text{CO}} / (2F \cdot n_{\text{site}})$$

Where I_{CO} is partial current for CO production; F is the Faraday constant ($96485.3 \text{ C mol}^{-1}$); n_{site} is number of active sites.

For atomically dispersed catalysts, each metal site is considered as a catalytic site, while for bulk metal catalysts, the number of active sites is determined by electrochemical methods.

The details of n_{site} evaluation in the cited literature are introduced below.

(1) $\text{Zn}_1\text{Ni}_4\text{-ZIF-8}$ (*Energy Environ. Sci.* **11**, 1204-1210 (2018)): TOF data was extracted from Fig. 3d.

(2) $\text{Fe}^{3+}\text{-N-C}$ (*Science* **364**, 1091-1094 (2019)): TOF data was extracted from Fig. 2d.

(3) OD-Au (*J. Am. Chem. Soc.* **134**, 19969–19972 (2012)): The electrochemical surface area of the oxide-derived Au electrode was determined by measuring the charge associated with the stripping of an underpotential deposited Cu monolayer (Fig. S4). It is assumed that the atomic density on the electrochemical surface is that on Au (111) facet and all surface atoms were active sites.

(4) np-Ag (*Nat. Commun.* **5**, 3242 (2014)). The CO partial current density was determined using Fig. 7 in Supplementary Materials, and the electrochemical surface area for per 1 cm² of the electrode was calculated as 2650 cm² using cyclic voltammetry (Fig. 8 in Supplementary Materials). TOF was determined by assuming Ag (111) facet as the active site.

(5) CoPc-2 (*Nat. Commun.* **10**, 3602 (2019)): The CO partial current density was extracted from Fig. 5a, and active site concentration was obtained from Co^{II}/Co^I redox wave from CV under argon atmosphere (Fig. 2).

(6) A-Ni-NSG (*Nat. Energy* **3**, 140-147 (2018)): TOF data was extracted from Fig. 3c.

(7) NiN₄/C-NH₂ (*Energy Environ. Sci.* **14**, 2349-2356 (2021)): The CO partial current densities in H-cell and flow cell were extracted from Fig. 2c and Fig. 3e, respectively. They assumed that the metals on the electrodes were atomically dispersed, and each site was regulated as a catalytic site. Active site concentration was estimated from catalyst loading versus metal concentration.

(8) Ni-CNC-1000 (*Angew. Chem. Int. Ed.* **61**, e202113918 (2022)): TOF data in H-cell was extracted from Fig. 3e, while the Faradaic efficiency and CO partial current density in the flow cell were extracted from Fig. 5c. They assumed that the metals on the electrodes were atomically dispersed, and each site was regulated as a catalytic site. Active site concentration was estimated from catalyst loading versus metal concentration.

(9) NiFe-DASC (*Nat. Commun.* **12**, 4088 (2021)): The CO partial current density was extracted from Fig. 3d. Assuming that the Fe-Ni atom pair is catalytic site, and site concentration was estimated from catalyst loading versus metal concentration on the electrodes.

(10) N₃NiPc-CNT (*Energy Environ. Sci.* **14**, 1544-1552 (2021)): TOF data in H-cell was extracted from Fig. 3d, and the Faradaic efficiency of CO and total current densities in the flow cell were extracted from Fig. 3h. Active site concentration was estimated from catalyst loading versus metal concentration on the GDE.

(11) CoN₄-CNT (*ACS Catal.* **12**, 2513–2521 (2022)): TOF data was extracted from Fig. 3e.

1.16 In situ-operando XAS

- Could the authors suggest further explanations to the almost no differences observed during reaction? If CO coordination occurs (as suggested by SR-IRAS measurements), variations in the XANES spectra (figure S15) should be noticeable, more probable at peak B (1s-4p_z transition). Same if H₂O coordination is proposed at -0.2 V, differences in the spectra should be expected. Please suggest explanation to these points.

Response: Thanks. To improve signal quality during in situ-operando XAFS testing, using high sample loading is necessary. However, it is important to note that only a portion of the catalytic sites react with reactants, which presents a challenge for accurately evaluating reaction processed using XAFS results that provide bulk average information. To overcome these limitations, surface-sensitive SR-IRAS testing was conducted to help understand the reaction mechanism. In addition, detailed XANES spectra have been included in the Supplementary Materials to zoomed-in details of the changes observed (Fig. R11). Discussions have also added to the revised manuscript to improve further clarification.

Fig. R11. (A) Ni K-edge XANES spectra of Ni-N-C collected at various potentials, (B) corresponding oscillation curves. (C-E) Zoomed-in details of features I-IV in A.

1.17 Data under inert atmosphere is suggested to be added in order to provide comparison. In addition, data after electrolysis should be incorporated (coming back to OCP for instance).
Response: Thanks. We have incorporated additional XAFS findings in the revised manuscript and extended upon the discussion.

Fig. R12. (A) Ni K-edge XANES spectra of Ni-N-C collected at various potentials in Ar-saturated 0.5 M KHCO_3 , corresponding (B) oscillation curves and (C) FT-EXAFS spectra.

Fig. R13. (A) Ni K-edge XANES spectra of Ni-N-C collected at OCP after electrolysis, together with ex situ and -0.8 V_{RHE} spectra, corresponding (B) oscillation curves and (C) FT-EXAFS spectra.

1.18 Suggested to include a zoom insert in figure S15 to clearly observe the variations discussed on text.
Response: Thanks. We have provided zoomed-in details of XANES spectra in Fig. S15. Please see Fig. R11 above.

1.19 References

References are reported in different manner (some cases as et al, other cases reporting all the authors), please refer to all in the same way.

Response: Thanks. We have corrected the reference manner.

Reviewer #2 (Remarks to the Author):

In this paper, the asymmetrically coordinated Ni single atom catalyst achieved higher turnover frequency than noble metal nanoparticles. It is noticeable in the CO₂RR mechanism that a well-known poisoning ligand, CO, functions as a promoter for releasing a CO₂RR product in a thermodynamically preferable way. Overall, this work will lead the interest of the related research fields after making a few revisions.

2.1. (i) According to the EXAFS description in page 6, line 177-182, and (ii) based on the suggested CO₂RR mechanism, the number of coordinated CO molecules should increase from -0.5 V_{RHE} to -0.8 V_{RHE}, while the opposite in Fig. 3E. I think the insets for -0.5 V_{RHE} and -0.8 V_{RHE} are changed with each other.

Response: Thanks. Based on the results from operando experiments, our findings suggest that Ni sites start to adsorb two CO starting at -0.5V_{RHE}. At more negative potentials, we observed a portion of single CO adsorption, which we attribute to low-valence Ni sites which originate from the very negative voltages induced reduction of some relatively unstable Ni sites in Ni-N-C. This speculation was firstly supported by the 1st derivative of XANES spectrum collected at -0.8 V_{RHE}, which showed the average valence state of Ni sites was reduced to +1.04 (Fig. R1). In situ SR-IRAS results of Ni²⁺Pc, which present a peak at approximately 2095 cm⁻¹ further support the above speculation (Fig. R2). To better convey our findings, we modified Figure 3E in the revised manuscript, and its updated version is presented in Fig. R3.

Fig. R1. 1st derivative data of Ni K-edge XANES for Ni-N-C at -0.8 V_{RHE}.

Fig. R2. In situ SR-IRAS spectra of NiPc collected in CO₂-saturated 0.5 M KHCO₃.

Fig. R3. Schematic illustration of the potential-induced structure changes of low-valence Ni sites in Ni-N-C.

2.2. In Fig. 4, the authors compare the thermodynamics of CO₂RR with and without the second CO molecule, according to the mechanism as *CO₂ → *COOH → *CO → * + CO.

What about the effect of the second CO on the initial adsorption of CO₂ molecule? As starting from bare (*) in the HER thermodynamics of Fig. 4B, the CO₂RR in Fig. 4A should be also drawn for a full catalytic cycle.

Response: Thanks. Based on Fig. R4, the second CO can create a significant energy barrier during the initial adsorption of the CO₂ molecule. We have provided DFT results for the *+CO₂ → *CO₂ → *COOH → *CO → * + CO reaction cycle (Fig. R5). We have also incorporated corresponding revisions and discussions in the revised materials.

Fig. R4. DFT calculated reaction free energy diagrams of NiN₂-2CO for CO₂RR.

Fig. R5. (A) DFT calculated reaction free energy diagrams of NiN₄, NiN₂, and NiN₂-CO for CO₂RR. (B) Free energy diagrams of NiN₄, NiN₂ and NiN₂-CO for HER. (C) Schematic illustration of the CO₂RR process over NiN₂ sites.

2.3. It is not necessary to include, but the expression in the line 208-209 can be improved by d-band center theory:

"Compared with NiN₄, Ni-N₂ displayed stronger interaction with molecular orbitals of CO, which obviously altered its electronic structure."

It can be understood by the higher d-band center of Ni-N₂ that can form stronger metal-CO bond than the lower-lying Ni d states of Ni-N₄.

Response: Thanks. We have modified the expression.

2.4. (minor) Fig. S17 and S18 should be improved for readers (e.g. resolution, etc.) and there are typos (aborb, unaborb -> adsorb, unadsorb). Please put the (+) signs on the Bader charges

in Table S6 for clarity.

Response: Thanks. We have modified these figures.

Reviewer #3 (Remarks to the Author):

Reviewer report for NCOMMS-22-49056-T

Asymmetric Dinitrogen-Coordinated Nickel Single-Atomic Sites for Efficient CO₂ Electroreduction

By Yuzhu Zhou et al.

This paper describes the synthesis of single atomic Ni sites with dinitrogen coordination, obtained by thermal treatment of graphitic carbon nitride under Ar and NH₃ environment. The catalyst is then used as electrocatalyst for CO₂ reduction (CO₂RR), placed on carbon paper and tested with H cell and flow cell configuration. A suite of operando techniques is then applied to understand the behaviour of the catalyst during CO₂RR. While the performance appear to be appealing, some improvement to the characterisation techniques, processing and interpretation is needed.

Queries:

3.1 g-C₃N₄ is notoriously insulating, and it is only widely used for photocatalysis due to photoactivity/conductivity (doi:10.1002/anie.201403375). Could the authors explain if the state of treated g-C₃N₄ still remains or pretty much carburised? I think it is worth checking the structural and electrical properties of the g-C₃N₄ and treated NC with and without the metal to see if there is a difference in the N-doped carbon support.

Response: Thanks. We first carried out TEM analysis to analyze the structural changes of g-C₃N₄ with and without the metal. TEM observation reveals a typical stacking structure for these samples (Fig. R1A,B). Previous work (*Angew. Chem. Int. Ed.* **51**, 9689-9692 (2012)) corroborated that DICY was first assembled to form laid graphitic carbon nitrate (g-C₃N₄) at low temperature, which served as a template to confirm and guide patches of aromatic carbon intermediate (derived from the calcination of glucose) condensation between the interlayer gaps. XPS coupled with elemental mapping analysis reveals the uniform distribution of g-C₃N₄ in both precursors (Fig. R1E,F and Fig. R2A,B). For Ni-N-C and NC, TEM observation reveals a uniform, wrinkled sheet-like structure (Fig. R1C,D). XPS results indicate that C atoms are primarily sp² type, indicating that the precursors have transformed into a N-doped graphene structure at high temperatures (Fig. R2C). This is consistent with our previous work, which shows that the g-C₃N₄ can be completely decomposed at 750 °C (*Nano Res.* **11**, 2217-2228 (2018)). Furthermore, N 1s XPS spectra of both Ni-N-C and NC show that they mainly contain graphitic, pyridinic, and pyrrolic N, further evidence the complete transformation of precursors (Fig. R2D). Additionally, it is found that compared with NC, both pyridinic and pyrrolic N peaks of Ni-N-C show a slight shift to higher energies, which indicate the partial electron transfer from N to Ni. These results clearly demonstrated the structural changes of the g-C₃N₄ and treated NC with and without the metal. We have added corresponding discussions in the revised manuscript.

Based on Nyquist plots analysis, it can be observed that all samples show comparable charge transfer resistance. This can be attributed to the formation of an electrically conducting carbon layer resulting from glucose carbonization between the g-C₃N₄ structure (*Angew. Chem. Int. Ed.* **51**, 9689-9692 (2012)), which is further supported by UV-Vis results (Fig. R3,4). Combined with the structure analysis, it can be concluded that the structure and electrical properties of N-doped carbon supports are analogous and not the main factors influencing the performance. The performance is primarily derived from the Ni single-atom sites.

Fig. R1. TEM images of (A) Ni-N-C, (B) NC, (C) Ni-N-C precursor, and (D) NC precursor. Elemental mappings of (E) Ni-N-C precursor and (F) NC precursor.

Fig. R2. XPS spectra of (A) C1s and (B) N 1s for precursors of NC and Ni-N-C . XPS spectra of (C) C1s and (D) N 1s for NC and Ni-N-C.

Fig. R3. Nyquist plots of NC, Ni-N-C, and their precursors measured at OCP in 0.5 M KHCO_3 .

Fig. R4. UV-Vis-NIR spectra of various samples.

3.2 Figure 1D: and the discussion at page 3: the Ni-N-C EXAFS is rather featureless, which is

expected of a highly disordered/defective local structure. It is suggested to change the naming of the EXAFS features because it can be confused with the sub-panel figure 1 numbers. Also, instead of peaks, probably “features” are more appropriate. For feature “A”, the proposed non-centrosymmetry of the Ni scatterer will cause 3d-4p mixing, which gives rise to the increased pre-edge feature “A” intensity (see e.g. doi:10.1021/bi900087w). The assessment that the Ni oxidation state is <2+ is supported.

Response: Thanks. We have enriched the discussion on Fig. 1D. In addition, we have modified the labeling and description.

Fig. R5. Ni K-edge XANES spectra of Ni-N-C, Ni foil, and NiPc.

3.3 Performance-wise, the catalyst appears to be excellent at this scale. I’m wondering what the standing of the catalyst is if Au and Ag catalysts are also included in the comparison, as the authors claim in page 4 “Ni-N-C outperformed most reported catalyst under same electrochemical conditions”. I’m wondering if energy-efficiency calculation, especially for the flow cell (as it is done with similar conditions with others), have been performed. I think Ag is relevant to compare as it is the incumbent catalyst that has shown large current deployment and much longer stability (see e.g. doi:10.1038/s41565-020-00823-x)

Response: Thanks. We have included Au and Ag as reference (Table R1). Next, we have evaluated the energy efficiency of Ni-N-C in a flow cell and the results are presented in Fig. R6. Our Ni-N-C catalyst exhibits significant efficiency, which is either comparable to or superior to previously reported catalysts.

Table R1. Activity comparison of Ni-N-C with recently reported catalysts at $-0.7 V_{RHE}$ (H cell).

Catalysts	Electrolyte	FE _{CO} (%)	j _{CO} (mA cm ⁻²)	Reference
Ni-N-C	0.5 M KHCO ₃	98.5	37.6	this work
Ni-CNC-1000	0.5 M KHCO ₃	95.0	3.2	(8)
N ₃ NiPc-CNT	0.5 M KHCO ₃	99.0	13.5	(9)
Ni-N ₄ /C-NH ₂	0.5 M KHCO ₃	92.0	30.8	(10)
A-Ni-NSG	0.5 M KHCO ₃	96.5	26.5	(11)
NiFe-DASC	0.5 M KHCO ₃	93.5	20.7	(12)
C-Zn ₁ Ni ₄ ZIF-8	0.5 M KHCO ₃	98.0	40.0	(13)
CoN ₄ -CNT	0.5 M KHCO ₃	95.5	40.6	(14)
Fe ₂ -N ₆ -C-o	0.5 M KHCO ₃	89.0	11.9	(15)
Co-N ₂	0.5 M KHCO ₃	89.5	22.4	(16)
CoTMAPc@CNT	0.5 M KHCO ₃	98.5	19.2	(17)
Ag-200nm NWA	0.5 M KHCO ₃	84.0	10.3	(18)
Au sputtered electrode	0.5 M KHCO ₃	70.0	3.3	(19)

Fig. R6. CEE comparison of Ni-N-C with reported catalysts.

3.4 Although some degree of catalytic activity stability is displayed in Figure 2E, a pertinent question to single-atom type catalyst is the inevitable surface reconstruction and agglomeration. With the high current/turnover, I think catalyst migration and agglomeration may be unavoidable. Could the authors comment on this aspect? There are some reports suggesting various operando techniques to track, (for example doi:10.1021/acs.nanolett.0c03475; doi:10.1021/acs.chemrev.2c00495; doi:10.1038/s41929-018-0182-6), I'm wondering if such approach has been attempted (for example, concurrent with the operando XAS shown in Figure 3).

Response: Thanks. The reconstruction of catalysts in electrochemical reactions has been frequently observed. Our group have reported the catalyst reconstruction in CO₂RR before (*Adv. Funct. Mater.* **30**, 2000407 (2020)). However, single-atom catalysts possessing unique isolated active sites are highly active in CO₂RR, even at very high current (*Science* **364**, 1091-1094 (2019)). In this work, since the CO₂-to CO conversion involve the use and generation of gases, its potential and signal stability are severely impacted, particularly under high current/TOF (*iScience* **23**, 101094 (2020)), making it challenging to use operando methods to study possible changes in the catalyst. Therefore, we conducted post-analysis of the Ni-N-C catalyst after subjecting it to high-current condition. As shown in Fig. R7, the XRD and HAADF-STEM analyses of the post-catalyst reveal that its structure remained single-atomic, and no obvious crystalline formed; this was validated by the absence of a diffraction peak for crystalline structure or cluster/particles.

Fig. R7. (A) XRD and (B) HAADF-STEM analysis of Ni-N-C after operation.

3.5 One follow-up question for the stability: why the flow cell stability is only limited to 18 hours?

Response: Thanks. It is noted that the long-term CO₂RR stability of Ni-N-C in 1.0 M KOH is limited due to the severe salt accumulation (*JACS* **143**, 3245-3255 (2021)). To date, almost all CO₂RR catalysts can not keep stable for a very long time in KOH. To overcome this issue, a

strategy that involves refreshing the electrolyte every 15 hours is explored in this study. The results indicate that Ni-N-C in 1.0 M KOH can function for 60 hours at a current density of approximately $450 \text{ mA cm}_{\text{geo}}^{-2}$ (Fig. R8). Furthermore, the FE_{CO} of Ni-N-C in KOH remains consistently high, exceeding $>98\%$. In neutral conditions, Ni-N-C also exhibited good stability, as demonstrated by its ability to maintain high performance over extended periods of time at current densities of approximately $330 \text{ mA cm}_{\text{geo}}^{-2}$ (GDE) and $40 \text{ mA cm}_{\text{geo}}^{-2}$ (H cell).

Fig. R8. (A) Chronoamperometry curve and FE_{CO} of Ni-N-C measured at $-0.5 V_{\text{RHE}}$ for over 30 hours in GDE with 1.0 M KOH as electrolyte. (B) Chronoamperometry curves and FE_{CO} of Ni-N-C measured at $-0.7 V_{\text{RHE}}$ for over 50 hours in both H cell and GDE with 0.5 M KHCO_3 as electrolyte, and $-0.6 V_{\text{RHE}}$ for 60 hours in GDE with 1.0 M KOH as electrolyte. Refresh means the refreshment of electrolyte at that time.

3.6 The assignment and interpretation of SR-IRAS peaks are not as unambiguous as expected. Ni-CO (monocarbonyl) is expected around 2118 cm^{-1} in cited ref 26 but is ascribed to 2030 cm^{-1} in the manuscript. Perhaps, a comparison with ^{13}CO isotope (and examination of peaks at slightly wider wavenumber range, say 1800 to 2800 cm^{-1}) may reveal more information and give a bit more certainty. Is it possible that the authors are looking at co-adsorption of CO and OH or H? (See e.g. DOI: 10.1016/0039-6028(94)00744-6 and DOI: 10.1016/S0167-2991(96)80058-9). I am also curious if any in-situ measurement for the alternative Ni-PC has been done, and if different behaviour of CO adsorption (or co-adsorption) is also seen on Ni-PC.

Response: Thanks. The peak at approximately 2030 cm^{-1} is most likely attributed to single CO adsorption over low-valence Ni sites. To investigate this, we conducted a ^{13}CO isotope experiment. Unfortunately, this experiment did not yield any valuable information, which we suspect was due to the lower CO concentration in the electrolyte compared to CO_2 . Additionally, the weaker adsorption of CO over Ni-N-C than metals may have contributed to the failure of the isotope experiment. To address this, we instead performed a $^{13}\text{CO}_2$ isotope experiment. In Fig. R9, a broad peak ranging from 2255 - 2395 cm^{-1} which corresponds to the consumption of CO_2 and $^{13}\text{CO}_2$ was observed. As the potential was reduced to $-0.6 V_{\text{RHE}}$, we also detected other peaks that correspond to CO adsorption. However, due to the possibility of signal overlap between different CO adsorption peaks, we roughly assigned the detected peaks between 1960 - 2190 cm^{-1} . Our findings revealed an isotope shift of approximately 48 cm^{-1} , which is close to the reported value (*ACS Energy Lett.* **4**, 682-689 (2019)). Importantly,

these results are consistent with our previous SR-IRAS findings, which strengthens the validity of our research. Additionally, the presence of low-valence Ni sites can be supported by the 1st derivative of XANES spectrum collected at -0.8 V_{RHE}, which showed the average valence state of Ni sites was reduced to +1.04 (Fig. R10). It is supposed that low-valence Ni sites originate from the very negative voltages induced reduction of some relatively unstable Ni sites in Ni-N-C. The obtained SR-IRAS spectra of NiPc with a peak at approximately 2095 cm⁻¹ further suggest the above speculation (Fig. R11). While the broad peak made it challenging to distinguish between the adsorption types (single or co-adsorption), since NiPc mainly contains Ni-N₄ configuration, we can believe that only single CO adsorption occur over Ni sites in NiPc.

The H adsorption is difficult to detect in neutral conditions due to its low concentration, therefore, no H adsorption was observed in our study. Nonetheless, we did detect broad peaks around 3500 cm⁻¹ (Fig. R12), which we attribute to OH-related adsorption. These peaks gradually decreased with reducing potentials, indicating competitive adsorption between CO₂/CO and H₂O in a neutral solution. We believe that low-concentration OH is not dominant in adsorption and, as such, emphasize H₂O adsorption at -0.2 V_{RHE} in the revised manuscript (Fig. R13).

Fig. R9. In situ isotope analysis for Ni-N-C in ¹³CO₂-purged 0.5M KHCO₃.

Fig. R10. 1st derivative data of Ni K-edge XANES for Ni-N-C at -0.8 V_{RHE}.

Fig. R11. In situ SR-IRAS spectra of NiPc collected in CO₂-saturated 0.5 M KHCO₃.

Fig. R12. Schematic illustration of the potential-induced structure changes of low-valence Ni sites in Ni-N-C.

Fig. R13. Schematic illustration of the potential-induced structure changes of low-valence Ni sites in Ni-N-C.

3.7 XAS, Page 6, “The analysis of XANES spectra at different potentials found that the peaks A and B slightly changed, indicating...” This labeling of “Peaks A and B” are very confusing and is easily mistaken for the earlier IR spectroscopy. The terminology is also confusing, because actually Figure 3C is the R space derivative obtained from data in Figure S15A, which qualifies as EXAFS for the energy range is sufficiently large (8320 to 8420) to cover the oscillations (all Ni XAS shown in this work has the same energy range). The interpretation of the EXAFS data also can be improved. For example, what is the significance of ca. 0.03 Å of first neighbour (Ni-N) radial distance change? By the way the correct atomic distances are the ones listed in Table S5. What are the error values for the fitting, and if 0.03 Å is within the error? How much elongation should we be expecting in Ni-N when OH or CO is adsorbed on Ni (e.g. can this be corroborated with the calculation)?

Response: Thanks. We have made several modifications to improve the accuracy and interpretation of our XANES and FT-EXAFS data. The peaks have been relabeled, and the terminology has been updated (Fig. R14). Corresponding discussions have been modified.

The present consensus is that the accuracy of interatomic distance determination by FT-EXAFS is between 0.01-0.001 Å (*Nature* **435**, 78-81 (2005)), with a difference of 0.03 Å being considered significant. The fitting data recorded in Table S5 has been thoroughly analyzed, taking into account the impacts of phase shift, thus presenting the correct atomic distances. Based on our previous discussion, it should be highlighted that under neutral conditions, H₂O adsorption, rather than OH, should be the dominant factor. To further demonstrate the elongation of Ni-N bonds, we have carried out additional calculations as shown in Table R2. It is obvious that the Ni-N bond length after H₂O adsorption is almost unchanged, while it increases after CO adsorption. However, H₂O adsorption brings Ni-O bond with a long length, while CO adsorption brings a Ni-C bond with short length. Given that XAFS provides an average result, these trends are consistent with in situ XAFS results.

Fig. R14. (A) Ni K-edge XANES spectra of Ni-N-C collected at various potentials, (B) corresponding oscillation curves. (C-E) Zoomed-in details of features I-IV in A.

Table R2. Activity comparison of Ni-N-C with recently reported catalysts at -0.7 V_{RHE} (H cell).

Adsorption species	None	H ₂ O	CO
Ni-N length	1.93	1.93	2.02
Ni-C/O length	/	2.06(Ni-O)	1.79(Ni-C)

REVIEWER COMMENTS

Reviewer #1 (Remarks to the Author):

The authors made important changes in the precedent article by taking into consideration the remarks from the reviewers and adding further explanations and data. In general, the discussion has been extended and it clarified some previous concerns. The integrity of the catalyst material is also further addressed. Overall, the methods and results will be of interest for the scientific community. There are just few further remarks:

1. The figure S6 (or Fig. R2) along with the results presented in table R1 are not clear. The reported edge energy values for Ni-N-C (red line) seems to appear closer to 8340 eV whereas the one for NiPc (blue line) seems to appear closer to 8335 eV. Are the edge energy values in the table correct or inversed? Please verify and indicate the demonstration/determination of the average valence of +1.37 of Ni in Ni-N-C.
2. Similar observation for figure S12. The current density values in S12.B are one order of magnitude lower than the ones in figure S12.A. Please verify.
3. In the text is indicated for Fig. S28: "However, when operando testing was performed in Ar-saturated electrolyte, the main peak corresponding to FT-EXAFS shifted significantly to the right only at -0.8 VRHE, which may be caused by the adsorption of oxygen-containing species". This shift is not really evident. Could you please include an arrow or a value to indicate where the significant shift is present?
4. Minor: the expression peak a, b and c for the operando SR-IRAS could be modified as writing them in between "" or in bold so the reader can differentiate it from the text avoiding confusion.

Reviewer #2 (Remarks to the Author):

The manuscript is now well displaying the comprehensive analyses on the electrochemical performances and theoretical understandings on the material, while some minor corrections are still remaining to be published.

1. Please check some typos and grammar again, for example,

(Page 3, Line 90) (and fig. S5) -> (fig. S5)

(Page 6, Line 179) maintain stable -> maintain stability

(Page 9, Line 248) evaluating reaction processed using XAFS -> evaluating reaction processes using XAFS

(Table S7) The title is the same with Table S8 and mislabelled.

(Page 9, Line 262) form *CO₂, then *CO₂ -> form *CO₂, and then *CO₂

2. Clarity in figures

(Page 4, Line 103-105) Based on Figure 1D, the ratio of feature III to IV is the maximum in NiPc, which is contradictory with the statement. I think the labels in Figure 1D is just mislabelled - red for Ni-N-C and blue for NiPc as done in fig. S6, also based on the consistency between Fig. 1D and fig. S6.

(Figure 4A) Please match the energy values well with the corresponding steps; for example, it would be more readable to color 0.72 eV in green, 0.41eV in blue and 1.15 eV in red, respectively.

(Figure 4C) The cyclic figure is a little bit confusing, since all of the steps involving those where (1, left) no CO adsorbed and (2, right) one CO adsorbed are connected and circulated. Please separate and rearrange them so that they start at different initial structures by clearly showing the initial structures. Otherwise, it is also clear to display the only pathway occurring in NiN₂-CO.

(Figure S31) For clarity, the DOSs should be sorted in a systematic way. For example, Ni-unads, Ni-ads, CO-unads, Co-ads; or Ni-unads, CO-unads, Ni-ads, CO-ads.

3. Reference

(Page 8, Line 219) our previous SR-IRAS findings(ref)

4. Main text

(Page 9, Line 268-269) "This analysis was inconsistent with the experimentally observed excellent activity"

>> This text should be now modified or removed, since the aforementioned experimental data already confirmed that "single CO adsorption optimizes the CO₂RR" in Page 8, Line 221-224.

(Page 9, Line 274-275) Not only this, but the first CO adsorption lowers d-band center of Ni so that the second CO adsorption is less preferred, which means that the second CO can be now released with reduced cost as shown in Figure 4A. Since the main issue is the second CO₂ adsorption and CO release, it is better to address this point.

(Page 10, Line 280-281) To say 0.72 eV as a significant energy barrier, it is the same with the cost of initial CO₂ adsorption in NiN₂-CO. Rather, it is enough to say a larger energy barrier than CO emission, NiN₂-2CO → NiN₂-CO + CO as indicated in the last step of Figure 4A in NiN₂-CO.

Reviewer #3 (Remarks to the Author):

Thank you for the authors for the additional experiments, especially on the isotope studies and additional EXAFS measurement to demonstrate the validity of the data. While most of my questions are answered, I have two follow up questions that will make the article ready for publication.

Additional questions:

(1) I missed the fact that the loading used for EXAFS/XANES measurements are ten times larger than the CO₂RR measurement. Also the loading for SEIRAS measurements are different too. Could the authors comment whether these differences in loading will affect the characterisation result? I think in one of the answers to the reviewer the authors have realised the possible difference between bulk technique (e.g. XANES) and surface technique (like XPS and SEIRAS), and one line should be added to acknowledge this possibility. Overall I still agree that if taken together the array of evidence supports the authors proposition.

(2) SR-IRAS: I think the SR-IRAS data from Ni-PC is not very conclusive because the single peak wavenumber is closer to peak (a) than the claimed single Ni-CO vibration in Ni-N-C sample (peak C). I agree that this is a tricky issue. What the authors can do is to look for references that demonstrate if Ni-(CO)₂ coordination is possible in Ni-PC, and really state the assumption limitation of the argument clearly.

REVIEWER COMMENTS

Reviewer #1 (Remarks to the Author):

The authors made important changes in the precedent article by taking into consideration the remarks from the reviewers and adding further explanations and data. In general, the discussion has been extended and it clarified some previous concerns. The integrity of the catalyst material is also further addressed. Overall, the methods and results will be of interest for the scientific community. There are just few further remarks:

1.1 The figure S6 (or Fig. R2) along with the results presented in table R1 are not clear. The reported edge energy values for Ni-N-C (red line) seems to appear closer to 8340 eV whereas the one for NiPc (blue line) seems to appear closer to 8335 eV. Are the edge energy values in the table correct or inversed? Please verify and indicate the demonstration/determination of the average valence of +1.37 of Ni in Ni-N-C.

Response: Thanks for pointing out this mistake. We inversed the labeling of Ni-N-C and NiPc. We have corrected the labeling of Ni-N-C and NiPc in Fig. S6 of the revised manuscript. The corrected figure is shown in Fig. R1 below.

Fig. R1. 1st derivative data of Ni K-edge XANES for Ni foil, Ni-N-C, and NiPc.

1.2 Similar observation for figure S12. The current density values in S12.B are one order of magnitude lower than the ones in figure S12.A. Please verify.

Response: Thanks. Fig. S12B should exhibit the partial current density and Faradaic efficiency of CO (j_{CO} and FE_{CO}). Based on the low FE_{CO} , j_{CO} in Fig. S12B exhibits very low values during measured potentials. Therefore, to make the figure more comprehensive, we have modified it as Fig. R2.

Fig. R2. Chronoamperometry curves (A) and FE_{CO}/j_{CO} (B) of NC measured at applied potentials in CO_2 -saturated 0.5 M $KHCO_3$.

1.3 In the text is indicated for Fig. S28: “However, when operando testing was performed in Ar-saturated electrolyte, the main peak corresponding to FT-EXAFS shifted significantly to the right only at -0.8 VRHE, which may be caused by the adsorption of oxygen-containing species”. This shift is not really evident. Could you please include an arrow or a value to indicate where the significant shift is present?

Response: Thanks. We have provided the zoom-in figure to emphasize the shifts in the revised manuscript (Fig. R3).

Fig. R3. (A) Ni K-edge XANES spectra of Ni-N-C collected at various potentials in Ar-saturated 0.5 M KHCO_3 , corresponding (B) oscillation curves and (C) FT-EXAFS spectra.

1.4 Minor: the expression peak a, b and c for the operando SR-IRAS could be modified as writing them in between “” or in bold so the reader can differentiate it from the text avoiding confusion.

Response: Thanks. We have written them in bold in the revised manuscript.

Reviewer #2 (Remarks to the Author):

The manuscript is now well displaying the comprehensive analyses on the electrochemical performances and theoretical understandings on the material, while some minor corrections are still remaining to be published.

2.1 Please check some typos and grammar again, for example,

(Page 3, Line 90) (and fig. S5) -> (fig. S5)

(Page 6, Line 179) maintain stable -> maintain stability

(Page 9, Line 248) evaluating reaction processed using XAFS -> evaluating reaction processes using XAFS

(Table S7) The title is the same with Table S8 and mislabelled.

(Page 9, Line 262) form $\ast\text{CO}_2$, then $\ast\text{CO}_2$ -> form $\ast\text{CO}_2$, and then $\ast\text{CO}_2$

Response: Thanks. We have corrected these errors and carefully checked the manuscript again.

2.2 Clarity in figures

(Page 4, Line 103-105) Based on Figure 1D, the ratio of feature III to IV is the maximum in NiPc, which is contradictory with the statement. I think the labels in Figure 1D is just mislabelled - red for Ni-N-C and blue for NiPc as done in fig. S6, also based on the consistency between Fig. 1D and fig. S6.

(Figure 4A) Please match the energy values well with the corresponding steps; for example, it would be more readable to color 0.72 eV in green, 0.41 eV in blue and 1.15 eV in red, respectively.

(Figure 4C) The cyclic figure is a little bit confusing, since all of the steps involving those where (1, left) no CO adsorbed and (2, right) one CO adsorbed are connected and circulated. Please separate and rearrange them so that they start at different initial structures by clearly showing the initial structures. Otherwise, it is also clear to display the only pathway occurring in NiN₂-CO.

(Figure S31) For clarity, the DOSs should be sorted in a systematic way. For example, Ni-unads, Ni-ads, CO-unads, Co-ads; or Ni-unads, CO-unads, Ni-ads, CO-ads.

Response: Thanks. We inversed the labeling of Ni-N-C and NiPc in Fig. S6. In the revised manuscript, we have corrected the labeling of Ni-N-C and NiPc in Fig. S6. The corrected figure is shown in Fig. R1 below.

Fig. 1D shows that the ratio of feature III to IV of Ni-N-C is higher than that of NiPc, which is consistent with our discussion in the main text.

Fig. R1. 1st derivative data of Ni K-edge XANES for Ni foil, Ni-N-C, and NiPc.

In addition, we have changed the color of energy values in Fig. 4A to enhance its readability (Fig. R2) and Fig. 4C has been divided into two separate figures (Fig. R3). Fig. S31 has been modified systematically as well, and it is presented as Fig. R4.

Fig. R2. DFT calculated reaction free energy diagrams of NiN₄, NiN₂, and NiN₂-CO for CO₂RR.

Fig. R3. Schematic illustration of the CO₂RR process over NiN₂ (C) and NiN₂-CO (D) sites.

Fig. R4. Projected density of states of NiN₂ and NiN₂-CO before and after CO adsorption.

2.3 Reference

(Page 8, Line 219) our previous SR-IRAS findings(ref)

Response: Thanks. It is not necessary to cite references as these findings are the SR-IRAS results mentioned previously in this work (Fig. 3A). For clarity, we put the corresponding figure in parentheses at the end of the sentence.

2.4 Main text

(Page 9, Line 268-269) "This analysis was inconsistent with the experimentally observed excellent activity"

>> This text should be now modified or removed, since the aforementioned experimental data already confirmed that "single CO adsorption optimizes the CO₂RR" in Page 8, Line 221-224.

(Page 9, Line 274-275) Not only this, but the first CO adsorption lowers d-band center of Ni so that the second CO adsorption is less preferred, which means that the second CO can be now released with reduced cost as shown in Figure 4A. Since the main issue is the second CO₂ adsorption and CO release, it is better to address this point.

(Page 10, Line 280-281) To say 0.72 eV as a significant energy barrier, it is the same with the cost of initial CO₂ adsorption in NiN₂-CO. Rather, it is enough to say a larger energy barrier than CO emission, NiN₂-2CO -> NiN₂-CO +CO as indicated in the last step of Figure 4A in NiN₂-CO.

Response: Thanks. We have modified related discussions in the revised manuscript.

Reviewer #3 (Remarks to the Author):

Thank you for the authors for the additional experiments, especially on the isotope studies and additional EXAFS measurement to demonstrate the validity of the data. While most of my questions are answered, I have two follow up questions that will make the article ready for publication.

3.1 I missed the fact that the loading used for EXAFS/XANES measurements are ten times larger than the CO₂RR measurement. Also the loading for SEIRAS measurements are different too. Could the authors comment whether these differences in loading will affect the characterisation result? I think in one of the answers to the reviewer the authors have realised the possible difference between bulk technique (e.g. XANES) and surface technique (like XPS and SEIRAS), and one line should be added to acknowledge this possibility. Overall I still agree that if taken together the array of evidence supports the authors proposition.

Response: Thanks. In response to the issue of loading used for bulk technique (XAFS) causing signal differences, we agree that this may bring some qualitative findings, however quantitative analysis is difficult. Therefore, to obtain a more sound conclusion, analysis exploring catalytic surfaces, such as SR-IRAS used in our work, is necessary. The influence from the catalyst loading should be smaller in surface-sensitive SR-IRAS than in XAFS. We have added an additional statement to acknowledge this possibility.

3.2 SR-IRAS: I think the SR-IRAS data from Ni-PC is not very conclusive because the single peak wavenumber is closer to peak (a) than the claimed single Ni-CO vibration in Ni-N-C sample (peak C). I agree that this is a tricky issue. What the authors can do is to look for references that demonstrate if Ni-(CO)₂ coordination is possible in Ni-PC, and really state the assumption limitation of the argument clearly.

Response: Thanks. Based on the newly cited reference, we have made a supplementary discussion to state the assumption of CO single adsorption on NiPc.